# CTCF couples long-range loop extrusion and diffusion to mediate a diverse *Igκ* repertoire

Emma L. Bush[1,4], Brigette Berke-Reynolds[1,4], Kaitlyn M. Hutchins [1], Xinrui Yu [2], Jorge A. Colón-Rosado [3], Fujung Chang[1], John Curran [1], Jiaxin Yang[2], Liangliang Sun [3], Jianrong Wang [2] ✉ & Yu Zhang [1] ✉

Dynamic genome folding is important for V(D)J recombination at the immunoglobulin kappa (*Igκ*) locus, which recombines Jκ and Vκ gene segments across a 3.2 Mb region in both deletional and inversional orientations. Chromatin loop extrusion and diffusion are considered two key mechanisms underlying *Igκ* locus folding, but how they coordinate remains unclear. Here we show that CTCF is a key regulator coupling loop extrusion and diffusion during *Igκ* V-J rearrangement, promoting recombination in both orientations across long genomic distances. Mechanistically, the CTCF N-terminus promotes long-range loop extrusion that facilitates distal Vκ usage by stabilizing cohesin against WAPL release, and also forms loop barriers enabling chromatin diffusion for inversional Vκ joining. In CTCF N-terminal-deficient B cells, defects in inversional Vκ joining are not restored by WAPL depletion but are instead largely rescued by a dCas9-blockade targeted to the Vκ-Jκ intergenic region, mimicking the CTCF barrier. Our findings thus highlight how CTCF coordinates distinct genome-folding mechanisms through its dual roles in cohesin stabilization and extrusion barrier formation to ensure the generation of a diverse *Igκ* repertoire.

The dynamic spatial organization of mammalian genomes play critical roles in differentiation and development, serving as the structural foundation for coordinated genetic and epigenetic activities[1–15]. A prominent mechanism involved in folding the genome into functional units is chromatin loop extrusion, which employs the cohesin complex to linearly drive progressive loop formation and synapsis of functional elements during this process[16–26]. The architectural protein CTCF binds to the asymmetric CTCF sites[27,28] and interacts with cohesin through its N-terminus, functioning as a directional barrier for cohesin extrusion to generate loop boundaries[29–34]. Structural studies revealed that a Tyr-Asp-Phe (YDF) motif in the CTCF N-terminus is critical for this polar interaction[29,34]. The other key mechanism of genome folding is diffusional capture that occurs via stochastic interactions[35], which lacks orientation and is not considered to be efficient for long-range interactions[36,37].

A concrete example of how dynamic genome folding instructs cellular function is V(D)J recombination, a programmed somatic recombination process that occurs in developing lymphocytes and generates diverse antigen receptor repertoires essential for adaptive immunity[14]. This process is initiated in the G1 phase of cell cycle by the RAG endonuclease, which cleaves a pair of recombination signal sequences (RSSs) flanking the V, D, and J coding segments[38]. The cleavage occurs in a stepwise manner: RAG first binds to one RSS, then finds and captures a complementary partner RSS for subsequent cleavage[38]. Given that antigen receptor loci can span several megabases (Mb), mechanisms are needed to overcome the bias of genomic distance and promote efficient synapsis of widely separated RSSs. In this regard, studies revealed that loop extrusion promotes RAG scanning along the chromatin within loop domains to locate partner RSS in convergent orientation for deletional joining[7,39,40]. This mechanism

[1]Department of Microbiology, Genetics and Immunology, Michigan State University, East Lansing, MI, USA. [2]Department of Computational Mathematics, Science and Engineering, Michigan State University, East Lansing, MI, USA. [3]Department of Chemistry, Michigan State University, East Lansing, MI, USA. [4]These authors contributed equally: Emma L. Bush, Brigette Berke-Reynolds. ✉e-mail: wangj164@msu.edu; yzhang22@msu.edu

plays a fundamental role in mediating V(D)J recombination at the mouse immunoglobulin heavy (*Igh*) chain locus, initially facilitating the physiological deletional joining of $D_H$ to $J_H$ segments in pre-pro-B cells[40]. This is followed by deletional V-to-$DJ_H$ recombination in pro-B cells, where developmental depletion of the cohesin releaser WAPL[21,22,41–43] promotes extended loop extrusion, enabling the bypass of a number of CTCF barriers at the *Igh* locus and allowing for the use of distal $V_H$ segments across the entire 2.4 Mb $V_H$ region[44,45].

Immunoglobulin light (*Igκ, Igλ*) chain loci undergo recombination following assembly of the *Igh* locus to generate B cell receptors. Unlike the *Igh* locus, which contains only forward $V_H$ segments for deletional joining, the *Igκ* locus includes both forward and reverse Vκ segments that recombine efficiently in deletional and inversional orientations, respectively[46]. The presence of robust Vκ-RSSs has been demonstrated to be important to promote inversional Vκ joining, however, this phenomenon cannot be solely explained by linear loop extrusion and requires a diffusion-based mechanism[47]. How the two processes coordinate to promote a diverse *Igκ* repertoire is largely unknown.

The 103 functional Vκ segments are distributed across the 3.15 Mb 5' *Igκ* region, with reverse Vκ segments located in the middle and forward Vκ segments at both the very proximal and distal ends[46]

(Fig. 1a). Broad *Igκ* recombination potential is mainly regulated by the intronic (iEκ) and 3' (3'Eκ) enhancers located adjacent to the four functional Jκs, which together form a recombination center to recruit partner Vκ-RSSs for recombination[48–51]. In addition to the recombination center enhancers, the Vκ region contains two recently identified enhancers, E88 and E34, which modulate nearby Vκ gene segment usage[52,53]. Importantly, Vκs are insulated from the recombination center by the Cer and Sis elements interacting with the Vκ region and recombination center, respectively[47]. Each element contains two CTCF sites, with those in Cer oriented toward Vκs and those in Sis oriented toward the recombination center, together supporting a balanced Vκ repertoire[54,55] in a manner dependent on the orientation of these CTCF sites[56]. Additionally, multiple CTCF sites are spread throughout the Vκ region[46,57,58]. Conditional depletion of CTCF in pre-B cells skews Vκ repertoire toward the use of proximal Vκ segments[59]. These studies implicated CTCF as an important regulator of *Igκ* recombination, however, the underlying mechanisms of its action remain elusive.

The molecular basis of how distinct chromatin-folding processes cooperate to enable efficient *Igκ* recombination in both deletional and inversional orientations remains unknown. In this study, we identify a central role for CTCF in coordinating long-range chromatin loop

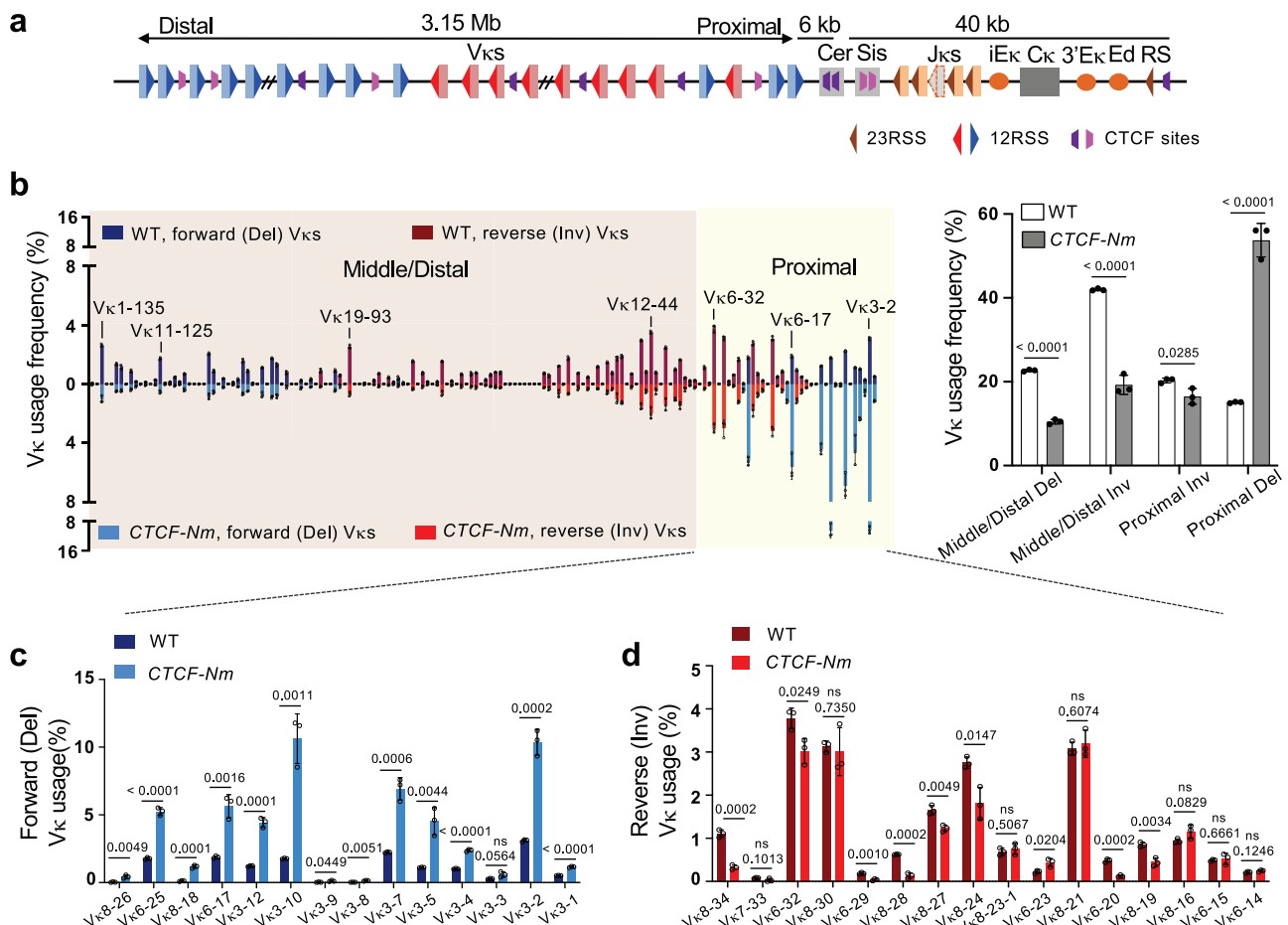

**Fig. 1 | *CTCF-Nm* mutation impairs distal Vκ utilization and exhibits differential effects on proximal deletional and inversional Vκ joining. a** Schematic of the murine *Igκ* locus (not to scale). Approximately100 Vκ segments span 3.15 Mb in both forward (blue, deletional-oriented) and reverse (red, inversional-oriented) orientations. A V-J intergenic region containing the Cer and Sis elements separates the Vκs from the downstream Jκs (orange) and constant (gray) region. Pink and purple trapezoids denote forward and reverse CTCF sites, and three enhancers (orange ovals) are positioned downstream of the Jκs. **b** Left, HTGTS-V(D)J-Seq analysis of Vκ usage frequency in WT (top) and *CTCF-Nm* (bottom) *v-Abl* pro-B cells.

Each Vκ utilization is shown as mean percentage of total VκJκ1 junctions ± SD from three independent replicates. Del stands for deletional joining and Inv stands for inversional joining. Right, summary of deletional and inversional Vκ usage in proximal versus middle/distal Vκ regions in WT and *CTCF-Nm* mutants. **c,d** Frequency of individual deletional Vκ usage (**c**) and inversional Vκ usage (**d**) in the indicated proximal Vκ region in WT and *CTCF-Nm* cells. Data represents mean ± SD from three independent replicates. All P values were calculated using unpaired, two-tailed Student's t test.

extrusion and short-range diffusion to balance recombination outcomes across the *Igκ* locus, through its dual functions in cohesin stabilization and loop barrier formation. These findings reveal that CTCF regulates immune diversity through tightly coupled yet distinct mechanisms and highlight its versatility in genome regulation, which may extend to other fundamental processes such as transcription and DNA repair.

## Results

### Multifaceted regulation of *Igκ* repertoire by CTCF N-terminus

To investigate the molecular basis of CTCF's role in *Igκ* recombination, we disrupted the CTCF N-terminal YDF motif critical for the polar interaction between CTCF and cohesin via the CTCF(Y226A/F228A) mutation[29] (thereafter referred to as CTCF-Nm). The homozygous *CTCF-Nm* was generated via CRISPR/Cas9 targeting[60] combined with a single-stranded DNA oligonucleotide (ssODN) template[61] in a RAG1 deficient *Eμ-Bcl2⁺ v-Abl* pro-B cell line[48] (Supplementary Fig. 1a, b). This cell line lacks endogenous RAG1 and carries a transgenic catalytically-dead RAG1(D708A)[48], which allows inducible V(D)J recombination[62] in G1-arrested cells upon wild-type (WT) RAG1 complementation. *v-Abl* pro-B cell lines have been demonstrated to be a reliable system to study *Igκ* rearrangement mechanisms[47]. We employed high-throughput genome-wide translocation sequencing-adapted V(D)J-sequencing (HTGTS-V(D)J-Seq)[7] with a Jκ1 bait to compare Vκ-to-Jκ joining profiles in WT and *CTCF-Nm* cells, which were RAG1-complemented and G1-arrested for 4 days.

WT cells underwent robust Vκ-to-Jκ1 recombination utilizing both forward and reverse Vκs across the 3.15-Mb Vκ region, generating deletional and inversional VκJκ joins, respectively (Fig. 1b, left panel). However, in the *CTCF-Nm* mutants, there was a considerable change in the pattern of Vκ utilization, with significantly decreased middle/distal Vκ usage accompanied by a substantial increase in the frequency of proximal Vκ usage (Fig. 1b, left panel). This suggests that the CTCF N-terminus plays an important role in supporting long-range Vκ joining, though it remains unclear at this point whether this effect occurs indirectly by suppressing proximal Vκ joining or directly by promoting distal Vκ joining. This pattern is similar to that reported for CTCF depletion[59], indicating CTCF N-terminus is essential for CTCF's function in *Igκ* recombination. We also noted a reduced level (3.7-fold) of absolute Vκ junctions in the *CTCF-Nm* mutants compared to WT cells, which might further indicate a negative impact on *Igκ* recombination center[48] activity or RAG activity by the *CTCF-Nm* mutation (Supplementary Fig. 1c).

Unexpectedly, while the mutation caused a consistent decrease in the usage of middle/distal Vκ segments regardless of Vκ orientation, it had a differential impact on proximal Vκs. We observed a 3.6-fold overall increase in the percentage of deletional, but not inversional, Vκ joins (Fig. 1b, right panel). Inspection of the utilization frequency of individual proximal Vκs showed a range of 2 to 15-fold increase in deletional Vκ usage (Fig. 1c), while inversional Vκ usage was either unchanged or reduced up to 5-fold (Fig. 1d). These observations strongly suggest a suppression of inversional Vκ joining in the *CTCF-Nm* mutants.

### Vκ-RSS orientation prescribes Vκ recombination activity in CTCF N-terminal mutants

We next investigated whether it is indeed the orientation, rather than the intrinsic sequence feature of Vκs, that contributes to the differential recombination potential in the *CTCF-Nm* mutant. To test this, we inverted a proximal Vκ region to see if the inversion, which reverses the Vκ orientation while maintaining the same sequence, would alter the joining frequency of the individual Vκ segments. To build the line, we first generated a *CTCF-Nm Igκ⁺/⁻* line harboring only a single *Igκ* allele by CRISPR/Cas9-mediated deletion of the entire *Igκ* locus on the other allele to facilitate the generation of mutant lines (Supplementary

Fig. 2a). The *CTCF-Nm Igκ⁺/⁻* line demonstrated very similar Vκ joining pattern across the *Igκ* locus with the parental line containing both alleles (Supplementary Fig. 2b). Next, on the intact *Igκ* allele, we inverted a 377-kb proximal Vκ region spanning from 6.9-kb upstream of Vκ8-34 to 7.3-kb downstream of Vκ6-15, which includes 4 forward Vκs and 15 reverse Vκs, to generate the *CTCF-Nm Igκ³⁷⁷ᵏᵇ⁻ⁱⁿᵛ* line (Fig. 2a, Supplementary Fig. 2c). Remarkably, upon this inversion, all 4 forward Vκs, now in reverse orientation for inversional Vκ-to-Jκ recombination, demonstrated greatly reduced joining activity with a 6 to > 28-fold decrease in the frequency of utilization (Fig. 2b, c). In contrast, most of the reverse Vκs (11 out of 15), now in forward orientation for deletional Vκ-to-Jκ recombination, showed significantly increased joining activity with a 2 to 39-fold gain in the frequency of utilization (Fig. 2b, d). These results strongly support the orientation of Vκ as a major determinant of proximal Vκ recombination potential in the context of CTCF N-terminal deficiency.

A potential limitation of the 377-kb inversion is that it spans a large chromosomal distance, which alters the relative linear proximity of the Vκ segments it contains to the Jκ segments. To minimize the impact of positional change, we also performed a localized mutation of the Vκ6-17 segment, in which only the 28-bp Vκ-RSS and its adjacent 10-bp coding region were inverted (Supplementary Fig. 3a, b). We found that reversing the orientation of this forward Vκ led to a substantial 9-fold decrease in its utilization frequency, from 5.58 to 0.64% (Fig. 2e, f, Supplementary Fig. 3c). This trend closely mirrors the one observed in the 377-kb inversion.

Collectively, the HTGTS-V(D)J-Seq analysis indicated two major roles of the CTCF-N terminus in *Igκ* recombination: (1) supporting long-range Vκ usage, and (2) promoting the utilization of inversional Vκs. We then sought out to explore the mechanistic basis.

### CTCF N-terminal mutation compromises loop regulation at *Igκ* locus

We hypothesized that the decreased utilization of middle and distal Vκs resulted from impaired long-range *Igκ* chromosomal interactions. To test this, we compared *Igκ* interaction profiles between WT and *CTCF-Nm* cells. We performed HTGTS-adapted chromosome-conformation-capture sequencing (3C-HTGTS)[63] with the Cer locale as bait in RAG deficient cells arrested in G1 for 4 days. The 3C-HTGTS analysis revealed that in WT cells, Cer interacted with genomic sequences across the Vκ region, with stronger interactions observed in the more proximal Vκ regions (Fig. 3a, Supplementary Fig. 4a). Most of the interaction peaks overlapped with CTCF or the E2A transcription factor[64,65] binding sites (BS) (Fig. 3a, Supplementary Fig. 4b). However, the *CTCF-Nm* mutation exhibited a significant reduction in long-range interactions with middle and distal Vκ regions, with this reduction affecting all peaks in these regions (Fig. 3a–c, Supplementary Fig. 4a, c). In the proximal Vκ region, mid-range interactions up to 670-kb were observed. However, while peaks at E2A BS were either maintained or increased, peaks at the CTCF BS were largely abolished in this region (Fig. 3a, c, Supplementary Fig. 4c). These findings suggest a dual role for the CTCF N-terminus in facilitating *Igκ* interactions: (1) mediating loop boundary formation for CTCF-dependent loops, and (2) promoting long-range interactions beyond this boundary function.

We wondered if the defective long-range *Igκ* loop formation in the *CTCF-Nm* cells resulted from compromised loop extrusion activity. Consistent with this hypothesis, Chromatin Immunoprecipitation Sequencing (ChIP-Seq) analysis of SMC3, a key component of the cohesin ring complex, showed a substantial reduction of cohesin accumulation on chromatin both genome-wide and at the *Igκ* locus in the *CTCF-Nm* cells compared to WT cells (Fig. 3d, e, Supplementary Fig. 5a), supporting that CTCF N-terminal deficiency may promote increased removal of cohesin from chromatin.

CTCF has been implicated in cohesin processivity by protecting cohesin from WAPL release[29,66]. To further investigate, we performed

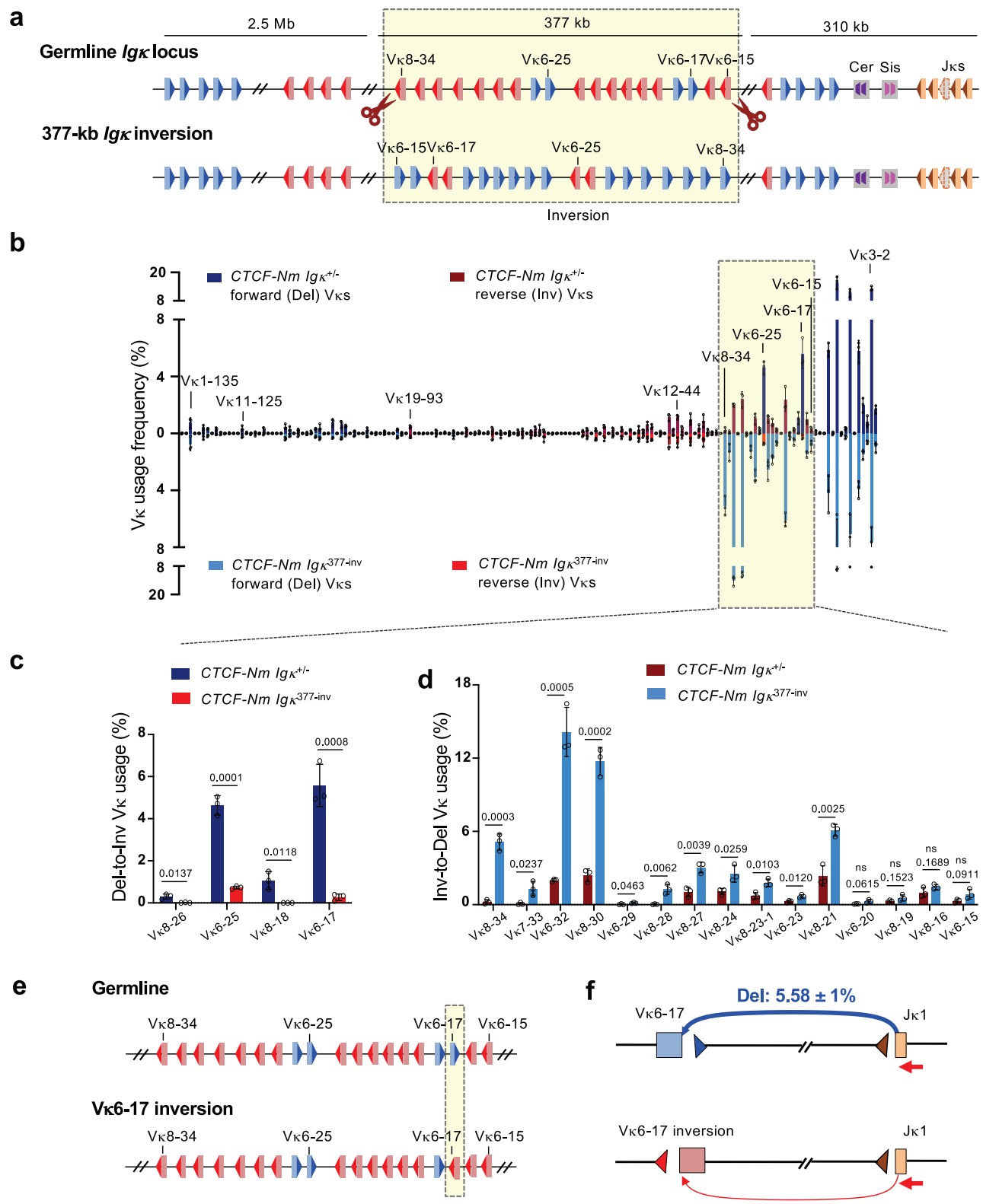

**Fig. 2 | Proximal Vκ recombination activity is largely determined by RSS orientation in the *CTCF-Nm* mutants. a** Schematic of the targeted inversion of the 377-kb proximal *Igκ* region in the *CTCF-Nm Igκ*[+/-] line. **b** HTGTS-V(D)J-Seq analysis of Vκ utilization frequency in *CTCF-Nm Igκ*[+/-] (top) and *CTCF-Nm Igκ*[377kb-inv] (bottom) cells. Data represent mean ± SD from three independent replicates. **c,d** Comparison of frequency of Vκ usage in the 377-kb proximal *Igκ* region in the *CTCF-Nm Igκ*[+/-] and

*CTCF-Nm Igκ*[377-inv] cells, analyzed for transition from deletional to inversional joins (**c**) or from inversional to deletional joins (**d**). Data represents mean ± SD from three independent replicates. P values were calculated using unpaired, two-tailed Student's t test. **e** Schematic of the targeted precise inversion of Vκ6-17 in *CTCF-Nm Igκ*[+/-] line. **f** Utilization frequency of Vκ6-17 in germline and Vκ6-17-inverted cells. Data represents mean ± SD from three independent replicates.

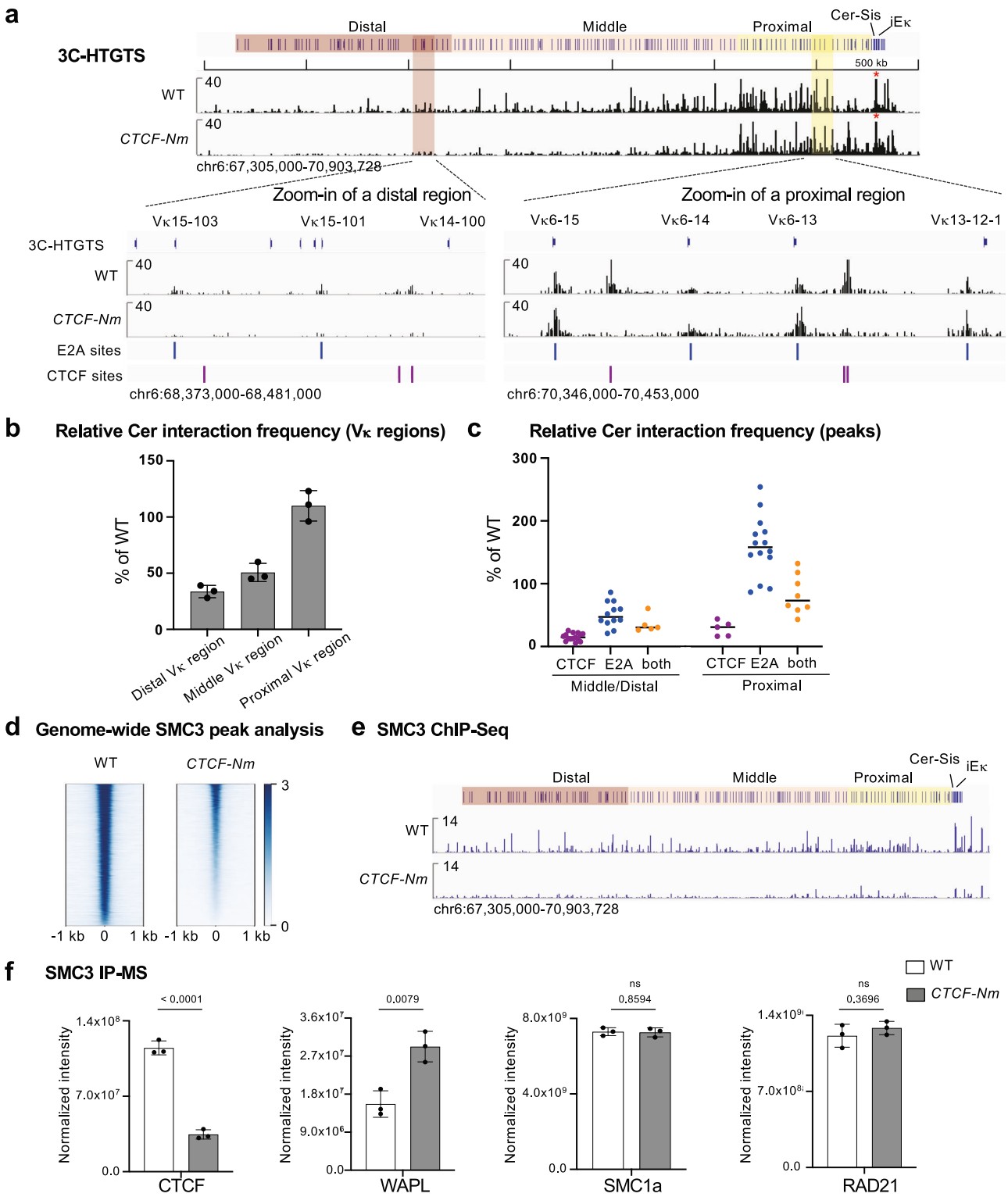

**Fig. 3 | CTCF N-terminus promotes long-range *Igκ* loop extrusion by protecting cohesin from WAPL release. a** Representative 3C-HTGTS analysis of *Igκ* chromatin interaction profile in WT and *CTCF-Nm* cells using Cer locale as bait. Bait region is indicated by a red star. Zoomed-in views show a representative distal Vκ region (lower left) and a proximal Vκ region (lower right), along with their correlation to E2A and CTCF binding sites. **b** Quantification of relative chromatin interactions. The relative Cer bait interactions in the indicated Vκ regions in *CTCF-Nm* cells are presented as a percentage of the WT values. Data represents mean ± SD from three independent replicates. **c** Analysis of peak interactions at the Vκ region. The relative Cer bait interactions of peaks overlapping with CTCF peaks (purple), E2A peaks (blue) or both (orange) in the indicated Vκ regions in *CTCF-Nm* cells are shown as a percentage of the WT values. Medians are denoted by the black lines. Each data point represents mean from three independent replicates. **d** Genome-wide peak analysis of SMC3 ChIP-Seq signal. Heatmaps show signal within ± 1.0 kb region across all peaks, ranked by SMC3 binding intensity. **e** SMC3 ChIP-Seq profiles at *Igκ* locus in WT and *CTCF-Nm* cells. **f** Quantitative mass spectrometry analysis of the relative abundance of SMC3 immunoprecipitates obtained from chromatin fraction of G1 arrested WT and *CTCF-Nm* cells. Protein intensities are normalized to the amount of chromatin-bound SMC3 across samples. Data represents mean ± SD from three independent replicates. P values were calculated using unpaired, two-tailed Student's t test.

label-free quantitative mass spectrometry (qMS) via nanoflow reversed-phase liquid chromatography (RPLC) on SMC3 immunoprecipitants from chromatin fraction of G1 arrested cells, normalizing to the chromatin-bound SMC3 for quantitative analysis. The IP-qMS results showed that the CTCF-Nm mutation led to a 3-fold reduction in the interaction between CTCF and SMC3, and notably, a 1.9-fold increase in the interaction between cohesin and WAPL (Fig. 3f). As a control, interactions between SMC3 and other core cohesin components, including SMC1a and RAD21, were shown to be unaffected (Fig. 3f). Chromatin loops are sensitive to changes in WAPL activity. It has been reported that a 2.2-fold increase in WAPL protein levels greatly alters genomic organization in pre-B cells compared to pro-B cells[67]. Therefore, the 1.9-fold increase in cohesin-WAPL interaction in *CTCF-Nm* cells is significant and likely accounts for decreased loop extrusion activity and the decreased long-range *Igκ* interactions observed in these cells.

We also noted that while the mutation only mildly reduced CTCF binding genome-wide and at the *Igκ* Cer-Sis elements, binding at Vκ locus was significantly decreased at many loci, suggesting many Vκ CTCF sites have relatively weak affinity that is dependent on a functional N-terminus of CTCF (Supplementary Fig. 5b). However, differences in CTCF binding among Vκ sites did not translate into differences in Vκ usage. For example, Vκs in a 400-kb distal region containing CTCF sites comparatively less affected by the mutation exhibited similar decreases in usage as other distal Vκs (Supplementary Fig. 5c). This suggests that although differential binding likely reflects intrinsic CTCF site strength, the binding retained in *CTCF-Nm* mutants may be non-functional due to the N-terminal mutation. In the proximal Vκ region, several inversional Vκs, such as Vκ6-32, Vκ8-30, Vκ8-21, Vκ8-16, and Vκ6-14, showed little change in usage upon *CTCF-Nm* mutation. To determine whether this was due to retained nearby CTCF binding, we examined the surrounding regions and found that these Vκs lack adjacent CTCF binding (Supplementary Fig. 5d). Instead, they were located near E2A sites, suggesting that retained or increased E2A-mediated interactions, rather than CTCF, may help preserve their usage in the mutant background. Finally, comparison of CTCF sites that retained versus lost binding in the *CTCF-Nm* mutants, both genome-wide and within the *Igκ* locus, did not reveal obvious differences in DNA sequence composition, implying that factors beyond the core CTCF motif and its immediate flanking sequences may contribute to N-terminal-dependent binding sensitivity (Supplementary Fig. 5e).

We further performed Precision Run-On sequencing (PRO-Seq) in WT and *CTCF-Nm* cells to assess whether the CTCF-Nm mutation affects expression of factors involved in loop extrusion[26] or diffusion[68–72]. This analysis revealed no significant changes in key genes for these processes (Supplementary Fig. 6a). Moreover, germline transcription across the Vκ region was largely preserved in *CTCF-Nm* mutants (Supplementary Fig. 6b), indicating that altered chromatin accessibility across the Vκ region does not explain the observed *Igκ* recombination defects in long-range Vκ usage and inversional joining. On the other hand, transcription across the recombination center, including the Jκ segments, iEκ, and 3'Eκ, was substantially reduced (Supplementary Fig. 6b), correlating with lower overall *Igκ* recombination and supporting a role for CTCF N-terminus in regulating recombination center activity (Supplementary Fig. 1c). Expression of V(D)J recombination factors was minimally affected, aside from enrichment of RAG1(D708A) and RAG2 that were transgenically expressed in these cells (Supplementary Fig. 6c), confirmed by RAG1 western blotting (Supplementary Fig. 6d). While retroviral complementation with abundant WT RAG1 may partly mitigate this effect for V(D)J recombination assays, potential WT heterodimerization with RAG1(D708A) may reduce fully active WT-WT dimers, contributing to lower absolute Vκ utilization alongside with reduced recombination center activity in *CTCF-Nm* cells.

Collectively, these findings support a direct role for the CTCF N-terminus in shaping the pattern of the *Igκ* repertoire by promoting loop barrier formation and protecting cohesin from WAPL release.

### Restoring long-range extrusion rescues distal deletional but not inversional joins

Given the increased cohesin removal by WAPL in *CTCF-Nm* mutants, we next asked whether depleting WAPL to enhance loop extrusion activity could rescue the defects in *Igκ* recombination. To test this, we first generated a line allowing for rapid degradation of WAPL using auxin-inducible degron (AID) technology. To minimize leaky degradation, we employed the AID version 2 (AID2) system[73]. In this system, we targeted *OsTIR1(F74G)* to the *Rosa26* locus on both alleles to ensure robust OsTIR1 expression (Fig. 4a, Supplementary Fig. 7a). WAPL was then tagged with AID2 and fused to tdTomato for inducible degradation with 5-Ph-IAA and tracking of WAPL level via flow cytometry analysis. We refer to this line as *Wapl-AID2*. Near complete WAPL degradation was achieved after 4 h of 1uM 5-Ph-IAA treatment and maintained during G1 arrest when 5-Ph-IAA was continuously present (Fig. 4b, Supplementary Fig. 7b, c). Finally, we introduced the CTCF(Y226A/F228A) mutation into the *Wapl-AID2* line to generate the *Wapl-AID2 CTCF-Nm* line. Comparative analyses were then conducted between untreated and WAPL-depleted cells using this line.

3C-HTGTS analysis using the Cer bait showed that as expected, untreated cells exhibited an *Igκ* interaction profile similar to that of *CTCF-Nm* cells, with the majority of interactions occurring in the proximal Vκ region (Fig. 4c, d, Supplementary Fig. 7d). This was characterized by prominent peaks at E2A sites and lack of strong interaction peaks at CTCF sites (Supplementary Fig. 7e). On the other hand, WAPL-depleted *CTCF-Nm* cells showed significantly enhanced long-range interactions, extending into the middle and distal Vκ regions, as well as beyond the *Igκ* locus into neighboring domains, consistent with increased loop extrusion activity (Fig. 4c, d, Supplementary Fig. 7d). These changes were accompanied by reduced shorter-range interactions in the proximal Vκ region and, notably, the near-complete removal of interaction peaks at E2A sites seen in the untreated cells (Fig. 4c, d, Supplementary Fig. 7d, e). This suggests that E2A sites may act as weak barriers to loop extrusion, which can be readily bypassed when extended extrusion activity is enabled.

Next, we performed HTGTS-V(D)J-Seq analysis on RAG complemented cells to assess *Igκ* recombination. The untreated *Wapl-AID2 CTCF-Nm* cells exhibited a Vκ repertoire similar to that of the *CTCF-Nm* line, which is biased toward use of proximal deletional Vκ segments (Fig. 4e). Notably, WAPL depletion significantly increased the utilization frequency of distal deletional Vκ segments at the cost of proximal ones, a change closely correlated with alterations in genomic interactions within these regions (Fig. 4e). Comparable fold enrichment patterns of weak and strong distal Vκ-RSSs indicates that these changes result from increased loop-extrusion-mediated RAG scanning activity, rather than from selective targeting of certain strong Vκ-RSSs (Supplementary Fig. 7f). Unexpectedly, despite enhanced interactions within the middle/distal inversional Vκ region, there was no improvement in the utilization of middle/distal inversional Vκ segments (Fig. 4e). These results provide direct evidence that loop extrusion-mediated long-range interactions are adequate to mediate deletional, but not inversional, distal Vκ joining. Therefore, enabling diffusion that promotes orientation-independent interactions is necessary for inversional Vκ joining. We next investigated how this process is achieved.

### CTCF loop barrier function is required for inversional Vκ joining

Previous reports showed that WAPL depletion alone in pro- and pre-B cells has a mild impact on *Igκ* repertoire[45,67]. Our HTGTS-V(D)J-seq analysis revealed that treatment of *Wapl-AID2* cells with 5-Ph-IAA resulted in increased distal Vκ deletional joining and decreased

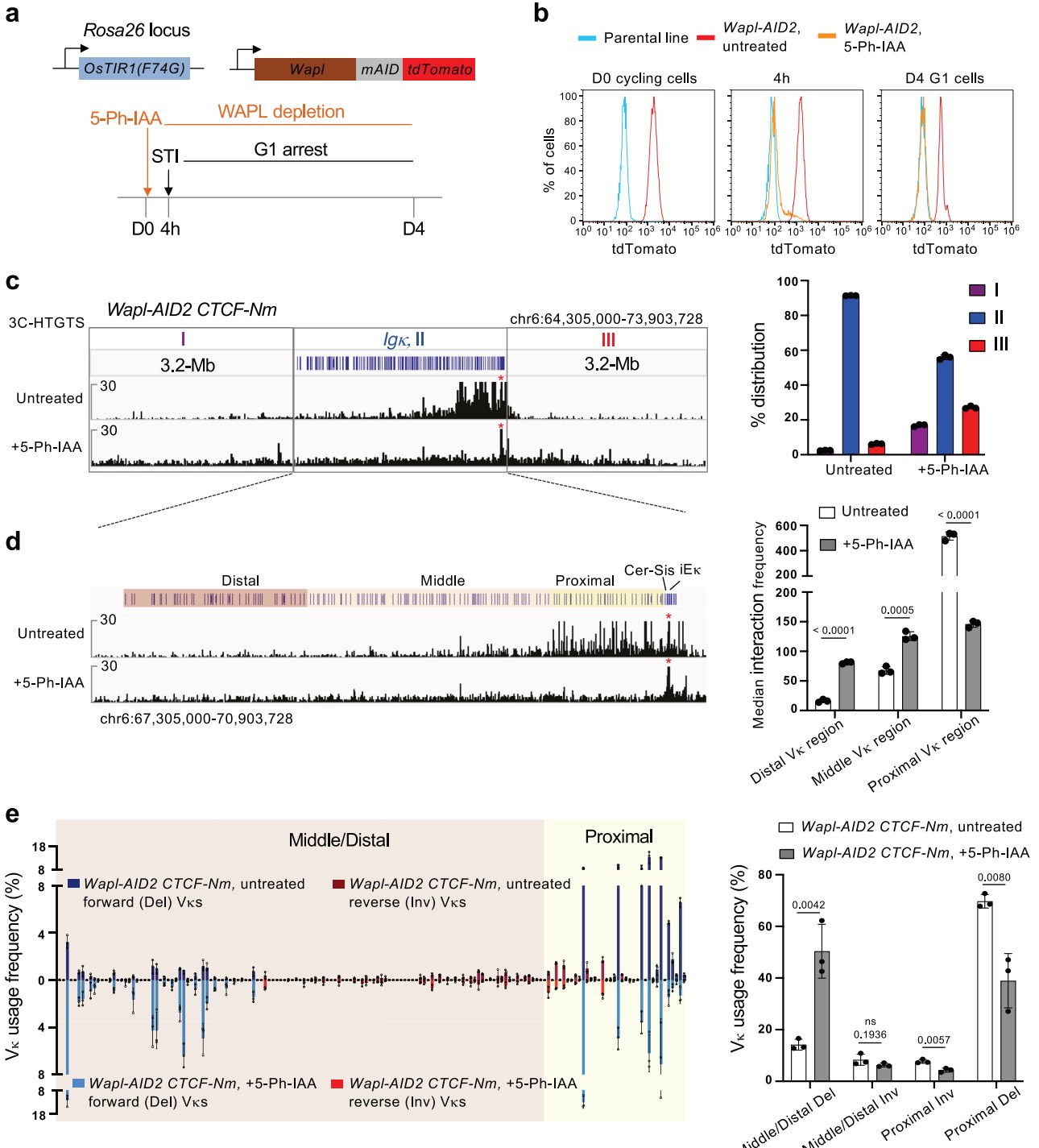

**Fig. 4 | WAPL depletion in *CTCF-Nm* cells rescues distal deletional, but not inversional Vκ joins. a** Top panel, diagram of the AID2 system for inducible degradation of WAPL. Bottom panel, schematic showing strategy of WAPL depletion in G1-arrested cells. Cells were first treated with 5-Ph-IAA for 4 h, followed by 4 days of STI-571 treatment to induce G1 arrest with continuous presence of 5-Ph-IAA. **b** Flow cytometry analysis demonstrating rapid degradation of WAPL after 4 h and persistent WAPL depletion in G1 cells upon 5-Ph-IAA treatment. The control parental line is the *Rosa26-OsTIR1(F74G) v-Abl* pro-B cell line. **c** Left, representative 3C-HTGTS analysis of untreated and 5-Ph-IAA treated *Wapl-AID2 CTCF-Nm* cells from Cer bait. I denotes the 3.2-Mb region upstream of the *Igκ* locus, II denotes the 3.2-Mb *Igκ* locus and III denotes the 3.2-Mb region downstream of the *Igκ* locus. Right, relative interactions of indicated regions are shown as percentage of the total

read counts within the region encompassing from 3.2-Mb upstream of *Igκ* to 3.2-Mb downstream of *Igκ*. Data represents mean ± SD from three independent replicates. **d** Left, zoomed-in views show *Igκ* interaction profiles in untreated and 5-Ph-IAA treated *Wapl-AID2 CTCF-Nm* cells. Right, quantification of interactions showing median interaction counts of indicated regions using 50-kb bins. Data represents mean ± SD from three independent replicates. P values were calculated using unpaired, two-tailed Student's t test. **e** Left, HTGTS-V(D)J-Seq analysis of Vκ usage frequency (%) in untreated (top) and 5-Ph-IAA treated *Wapl-AID2 CTCF-Nm* (bottom) *v-Abl* pro-B cells. Right, summary of deletional and inversional Vκ usage in proximal versus middle/distal Vκ regions in untreated and 5-Ph-IAA treated *Wapl-AID2 CTCF-Nm* cells. Data are shown as mean ± SD from three independent replicates. P values were calculated using unpaired, two-tailed Student's t test.

proximal Vκ joining, consistent with enhanced loop extrusion activity after WAPL depletion (Fig. 5a). On the other hand, most inversional Vκ joining was retained, in stark contrast to WAPL-depleted *CTCF-Nm* cells, which showed minimal inversional joining (Fig. 5a, b).

Unlike WAPL-depleted *CTCF-Nm* cells, cells with only WAPL depletion retain functional CTCF. We hypothesized that the loop barrier function of CTCF is critical to promote diffusion driving inversional Vκ joining. 3C-HTGTS with Cer bait showed that 5-Ph-IAA treated *Wapl-AID2* cells exhibited extended chromatin interactions, similar to WAPL-depleted *CTCF-Nm* cells (Fig. 5c, Supplementary Fig. 8a). However, unlike the WAPL-depleted *CTCF-Nm* cells that lacked detectable peaks in the Vκ region, 5-Ph-IAA-treated *Wapl-AID2* cells displayed multiple strong Cer-interaction peaks. Moreover, all of the interaction peaks detected within the Vκ region in 5-Ph-IAA-treated *Wapl-AID2* cells overlapped with CTCF sites (Fig. 5d, Supplementary Fig. 8a, b). These results confirmed that WAPL-depleted cells preserved strong CTCF barriers, which is required for inversional Vκ joining, though it remained unclear if the CTCF barriers at the *Igκ* locus, specifically the Vκ-Jκ intergenic sites versus Vκ region CTCF sites, are of equal importance.

**Targeted dCas9-SunTag blockade rescues inversional Vκ joining**

The CTCF barriers likely enable diffusion by preventing direct extrusion between Jκs and Vκs, a process that would otherwise inhibit inversional RAG recombination. This hypothesis predicts that extrusion barriers at the Vκ-Jκ intergenic region are of critical importance. To directly assess it, we tested whether targeted roadblocks mimicking CTCF barriers could rescue inversional Vκ joining in *CTCF-Nm* cells. Previous studies have shown that DNA-bound nuclease-dead Cas9 (dCas9) proteins can simulate extrusion barriers both in vivo and in vitro[34,40]. Additionally, it has been reported that similar to CTCF, a dCas9-blockade acts as a polar barrier to cohesin, halting its translocation when cohesin encounters the proto-spacer adjacent motif (PAM)-proximal side[34]. We generated the dCas9-blockade system in the untreated *Wapl-AID2 CTCF-Nm* cells. One copy of the *Igκ* allele was further deleted in the cells to facilitate targeting. To generate an efficient blockade, we leveraged multiple dCas9 BS with the bulky dCas9-SunTag system[74]. We inserted a synthetic sequence with 38 consecutive dCas9 BS into the Vκ-Jκ intergenic region, close to the Sis regulatory element. Of these, 20 dCas9 BS were oriented toward the Vκ region, and 18 were oriented toward the Jκ region, mimicking the CTCF site orientations in the Cer and Sis elements, respectively (Fig. 6a). This line is referenced as BS-only line. Next, we introduced a doxycycline-inducible, modified dCas9-SunTag system into the BS-only cell line using PiggyBac vectors (Fig. 6a). This modified system retained all components of dCas9-SunTag except for the VP64 transcription activator to avoid affecting the local epigenetic environment. This line is termed the dCas9-SunTag blockade line.

We confirmed stable expression of dCas9 protein and gRNA in the dCas9-SunTag blockade line (Fig. 6b, c), and ChIP-qPCR analysis showed strong dCas9 binding at the targeted locus (Fig. 6d). Furthermore, 3C-HTGTS analysis with iEκ bait demonstrated the formation of a targeted dCas9-SunTag-mediated loop barrier, which was evidenced by significantly decreased bait interaction with Cer and Sis elements located upstream of the dCas9 BS, while interaction with regions adjacent to the BS within the newly formed loop was enhanced (Fig. 6e). Therefore, direct loop extrusion from Jκ to Vκ was largely blocked in the dCas9-blockade line, enabling diffusional access between Jκs and Vκs. We did not detect increased interactions at the BS itself, likely because the dCas9-SunTag complex sterically hindered cleavage by NlaIII, the restriction enzyme used in 3C-HTGTS. This interpretation is consistent with previous reports showing that dCas9 can block restriction enzyme digestion[75] (Supplementary Fig. 9a).

Both the BS-only and dCas9-SunTag blockade lines were complemented with RAG1 and subjected to G1 arrest for HTGTS-V(D)J-Seq

analysis. The BS-only line exhibited a Vκ joining pattern similar to that of *CTCF-Nm* cells, which was primarily characterized by proximal deletional Vκ joining (Fig. 6f). Remarkably, the dCas9-SunTag blockade targeted to the Vκ-Jκ intergenic region largely rescued the Vκ repertoire deficiency. In the dCas9-SunTag blockade line, we observed a reduction in proximal deletional Vκ usage and a significant increase in middle/distal inversional Vκ joining as well as an increase in distal deletional Vκ usage, ultimately resulting in a Vκ joining pattern more akin to that of WT cells (Fig. 6f, Supplementary Fig. 9b). Collectively, these findings provide strong evidence that extrusion barriers, particularly at the Vκ-Jκ intergenic region, are crucial for inversional Vκ joining via diffusion.

## Discussion

*Igκ* recombination is a major immune diversification process generating the light chain of antibodies. Understanding how this large locus efficiently utilizes both deletional and inversional Vκ segments to create a diverse antibody repertoire is of great interest. It is generally believed that this process involves both loop extrusion and diffusion, but the exact coordination of these mechanisms was unclear[47,67]. Our studies addressed this question by demonstrating the molecular basis of CTCF's multifaceted role in *Igκ* recombination (Fig. 7a). Specifically, CTCF through its N-terminal interaction with cohesin, enables long-range loop extrusion by stabilizing cohesin against WAPL release, thereby promoting Vκ utilization across the 3.15-Mb Vκ region (Fig. 7a, I-IV). In the absence of this mechanism, long *Igκ* loops are disrupted, and the Vκ repertoire shifts toward proximal Vκ usage. However, direct extrusion from Jκ to Vκ promotes deletional Vκ joining but suppresses inversional Vκ joining, requiring the extrusion barrier function of CTCF to arrest cohesin translocation and promote diffusional interactions between Jκs and Vκs that facilitate recombination in an orientation-independent manner (Fig. 7a, V-VII).

DNA displays subdiffusive motion, which limits the ability of distant loci to physically interact via passive diffusion[69,76]. At the *Igh* locus, only the J$_H$-proximal DQ52 accesses J$_H$s via a diffusion mechanism while all other more distal D$_H$s synapse with J$_H$s via loop extrusion-mediated interaction[14,40]. Such a distance-dependent effect of diffusion versus loop extrusion is also thought to play a role in the synapsis of promoters and enhancers, with the pairing of distant promoters and enhancers being more strongly dependent on cohesin activity[36,37,77,78]. In this regard, loop extrusion likely brings the widely separated Vκs and Jκs into close proximity, within a subdiffusive range, at which point diffusion takes over (Fig. 7a, VIII). This aligns with a recent elegant study showing that the Vκ region loops to the Cer element and the Jκ recombination center loops to the Sis element[47], a configuration likely enabled by the specific orientation organization of CTCF sites within these elements, with their N-termini facing Vκs and Jκs, respectively. By identifying CTCF as a key factor converging long-range loop extrusion and diffusion, we demonstrate a mechanistic foundation for how different genome folding mechanisms are coordinated to provide essential structural foundations.

A recent study observed that a three-fold increase in WAPL expression had little effect on *Igκ* recombination and chromosomal configuration in pre-B cells, raising the question of whether long-range loop extrusion is necessary for distal Vκ utilization[67]. Our data showed that long-range loop formation between Cer and distal Vκ region is readily detected in WT cells and is dependent on CTCF's WAPL antagonizing effect (Fig. 3). In *CTCF-Nm* cells that impair this mechanism, while mid-range interactions up to 670-kb are still readily detected, we observed a 1.9-fold increase in cohesin-WAPL interactions, resulting in the removal of cohesin from chromatin, reduced long-range loop formation, and a corresponding decrease in middle and distal Vκ utilization (Figs. 1, 3). These findings suggest that this mechanism is critical for sustaining prolonged extrusion in a subset of cells to support long-range Vκ-to-Jκ recombination. Our data also

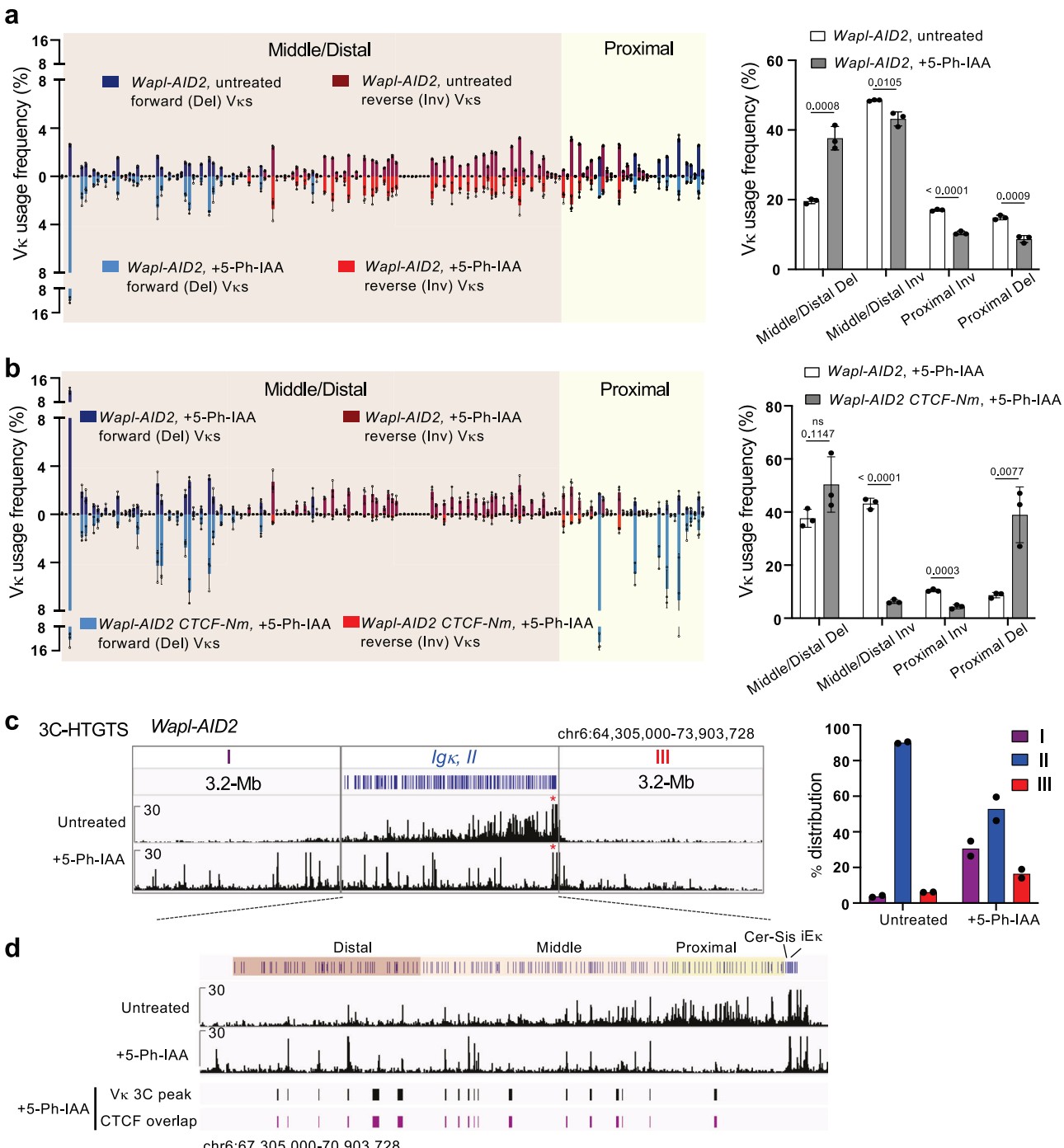

**Fig. 5 | WAPL depletion alone retains strong CTCF barriers and has little impact on inversional Vκ joining. a** Left, HTGTS-V(D)J-Seq analysis of Vκ usage frequency (%) in untreated (top) and 5-Ph-IAA treated (bottom) *Wapl-AID2* cells. Right, summary of deletional and inversional Vκ usage in proximal versus middle/distal Vκ regions in untreated and 5-Ph-IAA treated *Wapl-AID2* cells. Data are shown as mean ± SD from three independent replicates. P values were calculated using unpaired, two-tailed Student's t test. **b** Left, comparative HTGTS-V(D)J-Seq analysis of Vκ usage frequency in 5-Ph-IAA treated *Wapl-AID2* (top) and *Wapl-AID2 CTCF-Nm* (bottom) cells. Right, summary of deletional and inversional Vκ usage in proximal versus middle/distal Vκ regions in 5-Ph-IAA treated *Wapl-AID2* and *Wapl-AID2 CTCF-Nm* cells. Data are shown as mean ± SD from three independent replicates. P values

were calculated using unpaired, two-tailed Student's t test. **c** Left, representative 3C-HTGTS analysis of untreated and 5-Ph-IAA treated *Wapl-AID2* cells from Cer bait. Right, relative interactions of indicated regions are shown as percentage of the total read counts within the region encompassing from 3.2-Mb upstream of *Igκ* to 3.2-Mb downstream of *Igκ*. Data presents mean from two independent replicates. **d** Left, zoomed-in views show *Igκ* interaction profiles in untreated and 5-Ph-IAA treated *Wapl-AID2* cells. Locations of bait interaction peaks at Vκ region are indicated in black bars and interaction peaks that also overlap with CTCF binding sites are marked by purple bars. Only robust interaction peaks reproducible across replicates are included.

suggest a potential explanation for the observed WAPL insensitivity in pre-B cells: by competing with WAPL for cohesin binding, CTCF may stabilize cohesin at loop boundaries even in the presence of varying WAPL expression levels. Additionally, this mechanism explains the

CTCF site-based Cer element being a critical element regulating *Igκ* locus contraction[54,55].

Our findings also highlight that long-range loop extrusion is necessary but not sufficient for recombination of most inversional Vκs,

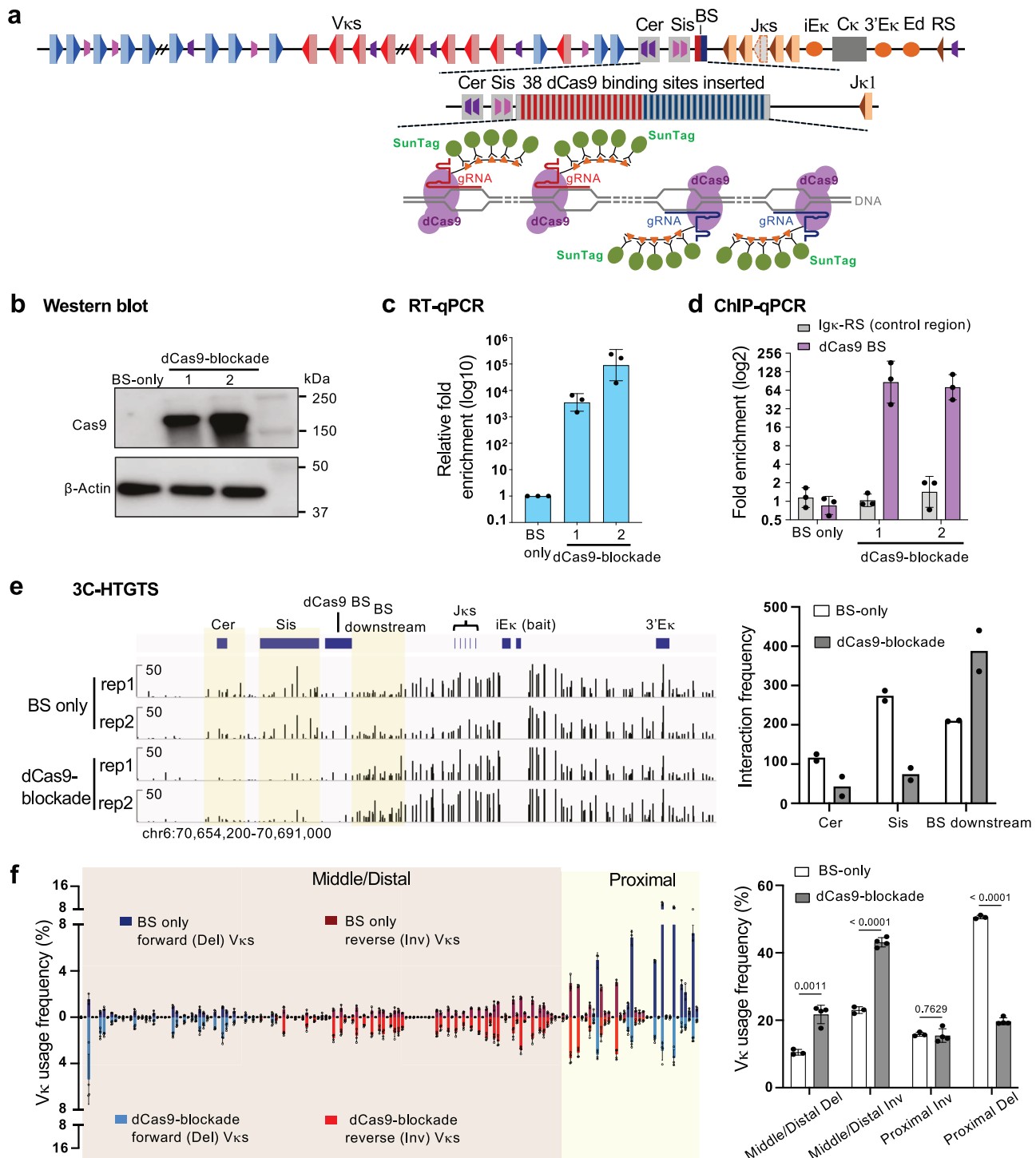

**Fig. 6 | Targeted dCas9-SunTag blockade rescues inversional Vκ joins in WAPL-depleted *CTCF-Nm* cells.** (**a**) Illustration of the dCas9-SunTag blockade system. 38 binding sites (BS) were inserted downstream of the Sis element to provide multiple binding sites for the dCas9-SunTag system. (**b**) Western blot confirmation of dCas9 expression in the dCas9-SunTag blockade line, derived from the *Wapl-AID2 CTCF-Nm* line. The experiment was repeated twice with similar results. (**c**) Reverse transcription-quantitative PCR (RT-qPCR) analysis showing the fold enrichment of gRNA expression in the dCas9-SunTag blockade line, relative to the control BS-only line. Data are shown as mean ± SD from three independent replicates. (**d**) ChIP-qPCR analysis showing the fold enrichment of dCas9-SunTag binding at the BS in the dCas9-SunTag blockade line and the control BS-only line. A region near *Igκ*-RS

was used as a negative control region. Data are shown as mean ± SD from three independent replicates. (**e**) Left, 3C-HTGTS profiles demonstrating formation of loop barrier near dCas9 binding sites in the dCas9-SunTag blockade line. Right, Quantification of bait interactions. Libraries were size-normalized, and interaction signals were counted for the indicated regions. Data are shown as mean from two independent replicates. (**f**) Left, HTGTS-V(D)J-Seq analysis of Vκ usage frequency (%) in the BS-only line (top) and the dCas9-SunTag blockade line (bottom). Right, summary of deletional and inversional Vκ usage in proximal versus middle/distal Vκ regions in the BS-only and dCas9-SunTag blockade lines. Data are shown as mean ± SD from three and four independent replicates, respectively. P values were calculated using unpaired, two-tailed Student's t test.

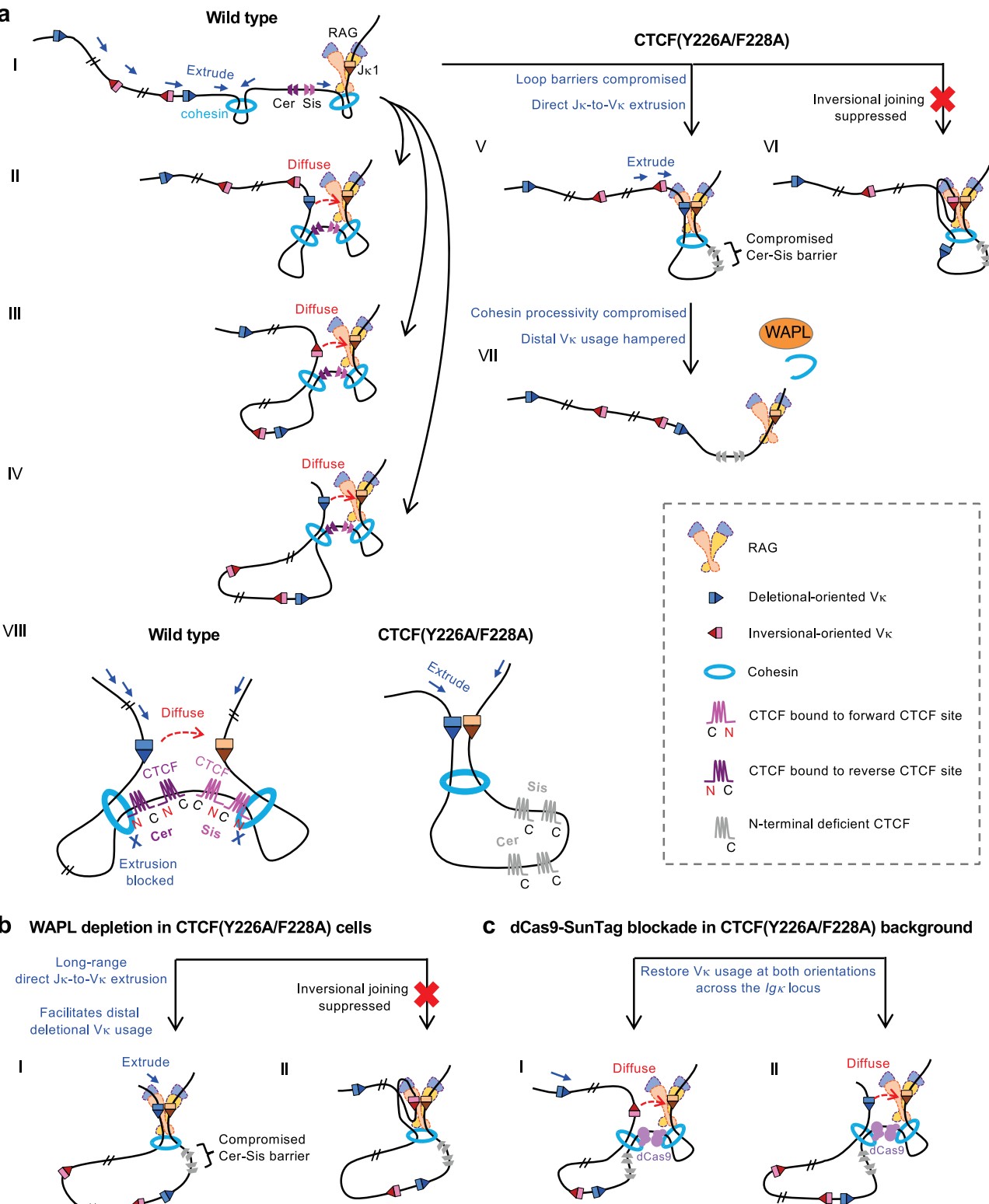

**Fig. 7 | Working models illustrating CTCF as a key factor converging long-range loop extrusion and diffusion to enable diverse Vκ recombination.** (**a**) Roles of the CTCF N-terminus in *Igκ* recombination. In wild-type cells, CTCF establishes loop barriers that facilitate diffusion, promoting Jκ joining with proximal Vκs in both deletional and inversional orientations. The most critical barriers are likely the Cer and Sis elements, which prevent direct Jκ-to-Vκ extrusion and promote short-range diffusion. For simplicity, only the deletional joining is illustrated for proximal Vκ region (I, II). Additionally, CTCF is important for cohesin processivity, which supports middle and distal Vκ joining (III, IV). In *CTCF-Nm* cells, CTCF cannot form effective loop barriers, resulting in Vκ joining occurring mainly through linear loop extrusion, which favors deletional joining (V) and suppresses inversional joining

(VI). Furthermore, frequent removal of cohesin by WAPL compromises middle and distal Vκ joining (VII). Panel VIII shows enlarged views illustrating how the CTCF N-terminus enables the Cer and Sis elements to function as polar barriers blocking extrusion from the Vκ and Jκ regions, respectively, thereby enforcing diffusion. This function is disrupted by the CTCF-Nm mutation, resulting in direct Vκ-to-Jκ extrusion. (**b**) Illustration showing that WAPL depletion in *CTCF-Nm* cells facilitates long-range direct Jκ-to-Vκ loop extrusion, rescuing distal deletional (I) but not inversional (II) Vκ joining. (**c**) Illustration showing that the dCas9-SunTag blockade targeted to the Vκ-Jκ intergenic region functions as a loop extrusion barrier, rescuing Vκ recombination in both deletional (I) and inversional (II) orientations across the locus.

which also depend on CTCF's role as an extrusion barrier. This contrasts with most deletional Vκ recombination that can occur efficiently without CTCF barriers. Accordingly, WAPL depletion in *CTCF-Nm* cells restores distal deletional Vκ joining but does not rescue inversional Vκ joining (Figs. 4e, 7b). Moreover, a striking difference was observed in inversional Vκ joining frequency between WAPL-depleted *CTCF-Nm* cells and WAPL-depleted only cells retaining CTCF barrier function (Fig. 5b).

Although long suspected, our study provides the direct evidence that mechanistically, the ability of CTCF barriers to block direct extrusion between Jκs and Vκs is essential for inversional Vκ joining as loop extrusion strongly favors deletional joins. In this regard, inversion of the proximal Vκ region in *CTCF-Nm* cells compromising CTCF barriers largely reverses the recombination strength of the contained Vκs based on their RSS orientations (Fig. 2b-d). Moreover, introducing a CTCF-barrier mimicking dCas9 blockade targeting the Vκ-to-Jκ intergenic region significantly enhances inversional Vκ joining across the *Igκ* locus in *CTCF-Nm* background (Fig. 6f).

The CTCF-Nm mutation substantially impaired CTCF binding at many Vκ sites, likely reflecting intrinsic site strength, but raised the question of whether reduced occupancy alone accounts for the *Igκ* recombination defect. In this regard, multiple observations argue instead for defective N-terminal-dependent stabilization of cohesin. Vκs within the distal Vκ region with relatively preserved binding still showed significantly reduced usage, indicating that the retained CTCF was non-functional (Supplementary Fig. 5c). WAPL depletion, which enhances cohesin processivity without known impact on CTCF binding, rescued distal deletional joining in the *CTCF-Nm* cells (Fig. 4e); and despite only mild reductions in binding at the Cer-Sis region, its long-range interactions with distal Vκ region, including those mediated by E2A, were substantially diminished (Supplementary Fig. 5b, Fig. 3a). Moreover, targeted dCas9 blockade near Cer-Sis, which is unlikely to affect Vκ CTCF binding, strongly rescued Vκ recombination across the locus (Fig. 6f). Together, these findings demonstrate that impaired CTCF-cohesin coupling, rather than reduced CTCF binding per se, underlies the *CTCF-Nm* phenotype.

The dCas9-blockade experiments underscore the significance of loop barriers in the Vκ-to-Jκ region (Fig. 7c). This raises the question of the role of other CTCF sites spread throughout the Vκ region. At the T cell receptor β locus, a CTCF site associated with the Trbv1 gene segment was shown to initiate loop extrusion from it and was proposed to act as an anchor to promote diffusional access of the nearby Vβ segment[79]. Further studies on mutations in Vκ-associated CTCF sites will be needed to determine whether they serve similar functions (Supplementary Fig. 10a). After the primary recombination between Vκ and Jκ1, which removes the Cer-Sis elements from the Vκ-Jκ region, *Igκ* can undergo secondary rearrangements in the case of non-productive rearrangement[80]. It's also possible that the CTCF sites in the Vκ region play significant roles in promoting Vκ joining during these secondary events (Supplementary Fig. 10b). In the absence of Vκ-Jκ intergenic barriers, these rearrangements are likely mediated predominantly by loop extrusion and can involve Vκ segments in forward germline orientation, as well as segments originally in reverse orientation that were flipped into forward orientation by an inversional primary recombination. Such inversional events can also reorient nearby reverse CTCF sites, thereby making them effective in facilitating local Vκ usage during secondary recombination (Supplementary Fig. 10b).

## Methods
### Cell lines
The parental RAG1(D708A), *Eμ-Bcl2 + v-Abl* pro-B cell line[48] was a gift from Dr. David Schatz. Parental and derivative *v-Abl* pro-B lines were maintained in RPMI 1640 medium (10-040-CV; Corning) supplemented with 10% FBS (Bio-techne, S11150H), 50 uM 2-Mercaptoethanol (Sigma-

Aldrich, M3148), 100 U/ml Penicillin-Streptomycin (Gibco, 15140122) and 2 mM L-Glutamine (Gibco, 25030081) at 37°C with 5% CO2.

### Generation of mutant *v-Abl* pro-B cell lines
CTCF(Y226A/F228A) mutation. We employed CRISPR/Cas9 targeting combined with a short single-stranded DNA oligonucleotide (ssODN) template[61] to generate the desired mutation. The ssODN (AAAAC-CAAAAAGAGCAAACTTCGTTACACAGAAGAGGGCAAA-GACGTGGATGTGTCTGTGGCCGATGCTGAGGAAGAACAGCAA-GAAGGACTGCTGTCTGAGGTTAATGCAGAGAAAGTAGTTGGTAAT) contained the target Y226A/F228A mutations and a silent mutation that disrupts the PAM motif of the gRNA targeting site. Cas9 protein (Integrated DNA Technologies, 1081059) and target-specific crRNA:-tracrRNA duplex were assembled as ribonucleoprotein and introduced into cells along with the ssODN via nucleofection using Lonza's 4D-Nucleofector system (SF solution, DN100 program). Cells were subcloned by dilution one day after nucleofection. Candidate clones were screened for homozygous mutation by PCR and validated by Sanger sequencing. Detailed oligo and primer information in this study are listed in Supplementary Table 1. All oligos and primers in this study were synthesized by Integrated DNA Technologies, unless otherwise specified.

*Igκ* allele modifications. The single *Igκ* allele mutation was generated via CRISPR/Cas9-mediated deletion of one entire 3.2-Mb *Igκ* allele, leaving the other allele intact. Candidate clones were screened by PCR and validated by Southern blotting. It served as the parental line for *Igκ* inversional mutations. The 377-kb proximal *Igκ*-inversion was made by inverting the region spanning from 6.9-kb upstream of Vκ8-34 to 7.3-kb downstream of Vκ6-15. Clones were validated by PCR and Southern blotting. The Vκ6-15 inversion was made by Cas9 aided targeting, along with an ssODN that contains the inverted Vκ6-15-RSS sequence and 10-bp flanking coding region. Clones were validated via PCR and Sanger sequencing.

WAPL-degron lines. Donor plasmid (pOsTIR1(F74G) -Rosa26) for *OsTIR1-V5* insertion was generated based on pEN113[18] (Addgene, 86233) using HiFi DNA Assembly kit (New England Biolabs, E2621S). The modifications included generating the *OsTIR1(F74G)* mutation and cloning homology arms for *Rosa26* locus integration. Upon Cas9 aided integration, cells were selected by 400 ug/ml G418 (Gibco, 11811-031) for 5 days and subcloned. Candidate clones were screened for homozygous mutation by PCR and validated by western blotting. The clones were then treated to remove the FRT-flanked NeoR-cassette by introducing pCAG-Flpo[81] (Addgene, 60662) carrying flippase to the cells to generate the *Rosa26-OsTIR1(F74G)* line. To further tag WAPL with mAID-tdTomato, pMK293[82] (Addgene, 72831) was modified to include homology arms flanking the WAPL stop codon and had mCherry replaced by tdTomato gene amplified from ptdTomato-N1[83] (Addgene, 54642) to generate pWAPL-mAID-tdTomato. Upon Cas9 aided targeting, tdTomato positive cells were subcloned via fluorescence-activated cell sorting and further validated by PCR, western blotting and flow cytometry analyzer.

dCas9-SunTag blockade lines. To generate the dCas9-SunTag blockade system in the untreated *Wapl-AID2 CTCF-Nm* cells, we first deleted one *Igκ* allele to facilitate analysis. To insert consecutive dCas9 BS to the Vκ-Jκ intergenic region, a donor plasmid (pgRNA1) was generated using the HiFi DNA Assembly kit. It harbors a 1710 bp synthetic sequence (synthesized by GenScript.) containing 38 repeats of a non-murine gRNA target sequence (GGTATGTCGGGAACCTCTGA) flanked by desired homology arms in the pBluescript KS(-) backbone. The synthetic sequence was verified by JASPAR vertebrata database[84] for lack of CTCF and E2A sites and inserted 376 bp downstream of Sis regulatory element via Cas9-aided integration to generate the BS-only line. The doxycycline inducible dCas9-SunTag and gRNA template were then introduced to the cells via PiggyBac vectors pPB-TRE3Gp-dCas9-SunTag-Puro and pPB-gRNA1-TetOn-3G-Activator-Neo,

respectively. The pPB-TRE3Gp-dCas9-SunTag-Puro vector was modified from pPB-Ins-TRE3Gp-dCas9-SunTag-VP64-EF1Ap-Puro[85] (Addgene, 183409) by deleting VP64. The pPB-gRNA1-TetOn-3G-Activator-Neo was modified from pPB-Ins-U6p-sgRNAentry-EF1Ap-TetOn3G-IRES-Neo[85] (Addgene, 183411) by cloning the gRNA template into the latter. Cells were selected by 400 ug/ml G418 (Gibco, 11811-031) and 2 ug/ml puromycin (Gibco, A1113803) for a week and then subcloned by limited dilution. Subclones were treated with 2 ug/ml doxycycline (Millipore Sigma, D9891) for 2 days and those with clear GFP induction were selected as positive clones for analysis. This line was then referenced as the dCas9-SunTag blockade line. Positive clones were further validated for dCas9 expression via western blot, gRNA expression via RT-qPCR, dCas9 binding via ChIP-qPCR and loop barrier formation via 3C-HTGTS. To induce dCas9-SunTag expression in G1 cells for analysis, 2 ug/ml doxycycline was added simultaneously with 3 uM STI-571 and again on Day 2 of G1 arrest.

## Chromatin fractionation and Co-IP

Isolation of chromatin-bound proteins and co-IP was performed based on a published protocol with slight modifications[86]. 40 million cells were lysed with cell lysis buffer (50 mM HEPES-KOH, pH 7.5, 140 mM NaCl, 1 mM EDTA, 10 % Glycerol, 0.5 % NP-0.4, 0.25 % Triton X-100, 1 M NaF, 1 M β-glycerophosphate, 1 mM dithiothreitol (DTT), 1X protease inhibitor cocktail (ThermoFisher Scientific, A32965)) on ice for 10 min, centrifuged to remove supernatant (cytoplasmic fraction) and resuspended in nuclei lysis buffer (10 mM Tris-HCl, pH 8, 200 mM NaCl, 1 mM EDTA, 0.5 mM EGTA, 1 M NaF, 1 M β-glycerophosphate, 1X protease inhibitor cocktail) on ice for 15 min. Further centrifugation was performed to remove supernatant (soluble nucleic fraction) and obtain the pellet of chromatin fraction. The chromatin fraction was solubilized via resuspending the pellet in Benzonase buffer (50 mM Tris-HCl, pH 7.5, 20 mM NaCl, 1 mM MgCl2, 0.2 % NP-40, 1 M NaF, 1 M β-glycerophosphate, 1X protease inhibitor cocktail, 50 U/ml Benzonase (Millipore Sigma, E1014)) and rotating at room temperature (RT) for 1 h. The supernatant was collected to measure the protein concentration, and an equal amount of total protein was used for the subsequent co-IP across all samples. During co-IP, the NaCl concentration of the chromatin fraction was adjusted to 100 mM. Samples were then pre-cleared with 20 ul Protein A Dynabeads (Invitrogen, 10001D) for 2 h at 4 °C and subjected to overnight incubation with 5 ug of SMC3 antibody (Fortis Life Sciences, A300-060A) or 5 ug of IgG isotype control (Invitrogen, 02-6102). The IP samples were captured by incubation with 40 ul Protein A beads for at least 2 h, washed five times with wash buffer (50 mM Tris-HCl, pH 7.5, 100 mM NaCl, 1 mM MgCl2, 0.2 % NP-40, 1 M NaF, 1 M β-glycerophosphate, 1X protease inhibitor cocktail), five times with PBS and then proceeded to label-free qMS.

## Label-Free qMS

Sample preparation. Sodium dodecyl sulfate (SDS) in 100 mM ammonium bicarbonate (ABC, pH = 8) was added to the protein-bound beads to achieve a final concentration of 2% (w/v). The sample was mixed thoroughly by pipetting, then sonicated, vortexed, and incubated at 90 °C for 10 minutes to release and denature the bound proteins. After centrifugation at 14,000 X g for 10 minutes at 14 °C, the supernatant was collected, and 1 μL of 500 mM DTT was added for reduction, followed by incubation at 95 °C for 15 minutes. Then, 2 μL of 500 mM iodoacetamide (IAA) was added for alkylation and incubated in the dark at room temperature for 20 minutes. The reaction was quenched with 1 μL of 500 mM DTT. SDS was removed, and proteins were digested using the SP3 method[87]. Briefly, 4 μL of a 1:1 mixture of carboxylate-modified hydrophilic (Sera-Mag, 45152105050250) and hydrophobic (Sera-Mag, 65152105050250) paramagnetic beads were

added to the sample, followed by the addition of acetonitrile (ACN, ~75% v/v). After incubation and bead separation, beads were washed with ACN, resuspended in 100 mM ABC, and digested with 50 ng trypsin overnight at 37 °C. Digestion was quenched with formic acid, and the peptide solutions were magnetically separated for Nanoflow RPLC-MS/MS analysis[88].

Nanoflow RPLC-MS/MS. Peptide separation prior to MS/MS analysis was performed using an EASY nanoLC-1200 (ThermoFisher Scientific) with a 75 μm i.d. × 20 cm C18 column. Buffer A contained 0.1% (v/v) FA, and buffer B consisted of 80% (v/v) ACN and 0.1% (v/v) FA. 1 μL of peptide sample was loaded with buffer A at a maximum pressure of 800 bar. Peptides were separated at 200 nL/min using a gradient: 2-8% (v/v) B in 5 min, 8-40% (v/v) B in 60 min, 40-80% (v/v) B in 5 min, and 80% (v/v) B for 10 min. MS/MS analysis was performed on a Q-Exactive HF mass spectrometer (ThermoFisher Scientific) with a full MS scan range of 300-1500 m/z, a resolution of 60,000, and AGC of 3E6. MS/MS resolution was 60,000 with AGC of 1E5 and a maximum injection time of 150 ms. The loop count was set to 15 (top 15), with a quadrupole isolation window of 2 m/z and normalized fragmentation energy of 28. The ion intensity threshold was 6.7E4, and dynamic exclusion was set to 15 s.

Protein identification was performed using MaxQuant 1.5.5.1[89] with the *Mus musculus* UniProt proteome database (UP000000589). Label-free quantification and match between runs were enabled with default parameters. Database search settings included Trypsin/P as the digestion enzyme, with oxidation of methionine and acetylation of lysine and protein N-termini as dynamic modifications, and carbamidomethylation of cysteine as a fixed modification. The maximum number of missed cleavages was set to 2. The first search peptide mass tolerance was set to 20 ppm, and the main search precursor mass tolerance to 4.5 ppm. A false discovery rate (FDR) of 1% was applied to proteins, peptides, and peptide-spectrum matches using the target-decoy approach. Samples include WT and *CTCF-Nm* SMC3 immunoprecipitants from three independent replicates, together with their IgG isotype controls. To determine the relative abundance of SMC3 immunoprecipitants, we used the protein intensity data from the database search and normalized them to the SMC3 level across the samples. As the *CTCF-Nm* cells contain the Y226A/F228A mutation, peptides with corresponding changes were included in the search and included for the analysis of CTCF intensity.

## ChIP-Seq library preparation and data analysis

IP and Input DNA for ChIP-Seq were prepared from G1 arrested, RAG1 deficient cells. 20 million cells were crosslinked in 10 ml prewarmed culture medium with 1% formaldehyde at RT for 10 min, then lysed on ice for 10 min in 1 ml cell lysis buffer (5 mM PIPES pH 8, 85 mM KCl, 0.5% NP-40) followed by 150 ul nuclei lysis buffer (50 mM Tris-Cl pH 8.1, 10 mM EDTA, 1% SDS) at RT for 10 min. 850 ul dilution buffer (16.7 mM Tris-HCl pH 8.1, 0.01% SDS, 1.1% Triton X-100, 1.2 mM EDTA, 167 mM NaCl) was added to chromatin, followed by sonication at 4 °C by Bioruptor (Diagenode, B01060001) (20 cycles of 30 sec on and 30 sec off). Lysates were precleared with 20 ul Dynabeads Protein A at 4 °C for 2 h. 1/30 of lysate was reserved as input, the remainder incubated overnight at 4 °C with 2.5 μg SMC3 (Fortis Life Sciences, A300-060A) or 2.5 μg CTCF (Millipore Sigma, 07-729) antibody, followed by capture with 40 ul Dynabeads Protein A for 2-4 h. Immunoprecipitates were washed sequentially in 1 ml each of cold dilution buffer, low salt wash buffer (0.1% SDS, 1% Triton X-100, 2 mM EDTA, 150 mM NaCl, 20 mM Tris-HCl pH 8.1), high salt wash buffer (0.1% SDS, 1% Trion X-100, 2 mM EDTA, 500 mM NaCl, 20 mM Tris-HCl pH8.1), LiCl wash buffer (0.25 M LiCl, 1% NP40, 1% NaDOC, 1 mM EDTA, 50 mM Tris-HCl pH 8.1) and 1X TE buffer (10 mM Tris-HCl pH 8.0, 1 mM EDTA), and eluted in elution buffer (100 mM NaHCO3, 1% SDS) at 65 °C for 25 min . Both IP and input DNA had the concentration of EDTA and NaCl adjusted to 0.1 mM and 0.2 M, respectively, and were de-crosslinked at

65 °C overnight and purified using MinElute PCR Purification Kit (Qiagen, 28004). ChIP-Seq libraries were prepared with NEBNext Ultra II DNA Library Prep Kit (New England Biolabs, E7645S) and sequenced via Illumina paired-end 150 bp sequencing. Samples were aligned to mm9 using Bowtie2, processed using SAMtools[90] v1.8, deduplicated using Picard (https://broadinstitute.github.io/picard/) and normalized using bamCoverage[91] with the --CPM flag. The E2A ChIPseq data was reanalyzed from public dataset (GSM546523)[65]. CTCF and E2A peak sites in the *Igκ* locus were called by MACS2[92] v2.11 narrow peak calling function using WT CTCF ChIP-Seq data in this study and the published E2A data, respectively. For genome-wide peak analysis, reference peak annotation was generated using MACS2 peak calling function on WT samples and retaining only peaks shared between replicates. Heatplots were generated using deepTools2[91] and excluded outliers that are larger than Q3 + 50xIQR (Q3: 75% percentile; IQR: Interquartile Range) of each column. To investigate potential differences in CTCF binding element sequences between WT and *CTCF-Nm* mutants, high-confidence peaks (MACS2 p-value < 0.05, further filtered by q-value < 0.01, and overlapping between replicates) were divided into two groups: (1) peaks shared between WT and mutant, and (2) peaks unique to WT. CTCF motifs within each peak were identified using a 19-bp core motif (MA0139.1) from the JASPAR vertebrata database. When multiple motif instances were present in a single peak, only the motif closest to the peak summit was retained for downstream analysis. Motif logos, including the core 19-bp motif plus 16 bp upstream and 18 bp downstream sequences, were generated using WebLogo[93] and compared between the two peak groups at both the genome-wide level and within the *Igκ* locus.

## WAPL depletion

Cell lines carrying the WAPL-degron system were treated with 1 uM 5-Ph-IAA (Bio Academia, 30-003). WAPL degradation was evaluated via flow cytometry analyzer comparing parental line, untreated and treated cells after 4 h. Cells were then G1 arrested by 3 uM STI-571 (Sigma-Aldrich, SML1027) with continuous presence of 1 uM 5-Ph-IAA for 4 days. WAPL degradation was checked again by flow cytometry analyzer with G1 arrested parental line and untreated cells as controls. Cells were then collected and subjected to various analysis as described below.

## HTGTS-V(D)J-seq library preparation and data analysis

To prepare for HTGTS-V(D)J-Seq, RAG1 was reconstituted in RAG1 deficient *Eμ-Bcl2 v-Abl* cells via retroviral infection of the pMSCV-RAG1-IRES-Bsr plasmid (a gift from Dr. David Schatz), followed by 4 days of 50 ug/ml blasticidin (Gibco, A1113903) selection to enrich for cells with virus integration. Cells were then arrested by 3 uM STI-571 for 4 days, extracted for genomic DNA and prepared for HTGTS V(D)J-seq libraries[7,94]. 2–4 ug of DNA were sonicated using the Diagenode Bioruptor (two cycles of 20 s on and 60 s off) and linearly amplified by Phusion polymerase (Thermofisher Scientific, F530L) with a Jκ1 biotinylated primer (1 uM). Single-stranded products were purified with Dynabeads MyOne Streptavidin C1 beads (Invitrogen, 65002) and ligated to a bridge adaptor (2.5 uM) via on-beads ligation using T4 DNA ligase (New England Biolabs, M0202L) at 16 °C overnight. Adaptor-ligated products were amplified by nested PCR with indexed Jκ1 and adaptor primers, followed by tag PCR with P5-I5 and P7-I7 primers. PCR Products (400–700 bp) were gel-purified and sequenced on an Illumina MiSeq with 300-bp paired-end reads, unless otherwise specified. Sequenced libraries were processed via a published pipeline[94] (http://robinmeyers.github.io/transloc_pipeline/) with the following filters applied: f.unaligned=1, f.baitonly=1, f.uncut=G10, f.misprimed=L10, f.freqcut=1, f.largegap=G30, f.mapqual=1, f.breaksite=1, f.sequential=1 and f.repeatseq=1. Duplicate reads were then removed. Sequencing reads were aligned to the mm9 genome for strains containing WT *Igκ* allele. For strains with *Igκ* modifications, which include the 377-kb proximal *Igκ*-inversion, Vκ6-15 inversion and dCas9 BS insertion, reads were aligned to modified mm9 genomes incorporating the relative targeted mutations. To analyze the Vκ utilization pattern, relative frequency of individual Vκ usage was presented as a percentage, which was calculated by dividing the read count of each Vκ segment by the total Vκ usage in the library.

## 3C-HTGTS library preparation and data analysis

3C-HTGTS[63] was performed on G1 arrested, RAG1 deficient cells. Briefly, 10 million cells were crosslinked in 12 ml culture medium with 2% formaldehyde (RT, 10 min), quenched with 0.125 M glycine, and lysed on ice for 10 min in cell lysis buffer (50 mM Tris-HCl pH 7.5, 150 mM NaCl,

5 mM EDTA, 0.5% NP-40, 1% Triton X-100, 1X Protease inhibitor cocktail). Pelleted nuclei were treated sequentially with 0.3% SDS and 1.8% Triton X-100 (37 °C, 1 h each), digested overnight with NlaIII (700 U, New England Biolabs, R0125L), heat-inactivated at 65 °C for 20 min, and ligated under dilute conditions with T4 DNA ligase (100 U, New England Biolabs, M0202L) at 16 °C overnight. Ligated chromatin was de-crosslinked with Proteinase K (Millipore Sigma, 3115852001) at 65 °C overnight, treated with RNase A (Invitrogen, 12091021) at 37 °C for 1 h, purified by phenol-chloroform extraction, and resuspended in 200 μl 1 X TE. 4 ug DNA per sample was processed for Illumina library preparation using the HTGTS-V(D)J-Seq method with Cer or iEκ bait and sequenced via Illumina paired-end 300 bp sequencing. Libraries were processed as described for HTGTS-V(D)J-seq. Sequencing reads were aligned to the mm9 genome, or modified genome for strains with the dCas9 BS insertion. In addition, we filtered reads aligned to telomeric and satellite repeat regions to avoid potential PCR artifacts. We also removed bait region reads (chr6:706586310-70660396 for Cer bait in the mm9 genome, and chr6:70677069-70678882 for iEκ bait in the modified genome with dCas9 BS insertion) to minimize the impact of self-ligation on quantitative analysis. The libraries were then size-normalized to total junctions of the smallest library in the set of libraries for comparison. 3C-HTGTS peaks were called using published code[47] (https://github.com/Yyx2626/HTGTS_related) and only recurrent peaks called in at least two independent replicates were kept for analysis.

## PRO-Seq library preparation and data analysis

PRO-Seq libraries were generated from G1 arrested, RAG1 deficient cells following a published protocol[95] with slight modifications. 5 million nuclei were collected by sucrose gradient centrifugation. Nuclear run-on was performed at 37 °C for 5 min by mixing nuclei with 2 X run-on master mix (10 mM Tris-HCl pH 8.0, 5 mM MgCl$_2$, 1 mM DTT, 300 mM KCl, 50 uM Biotin-11-CTP (Revvity, NEL542001EA), 0.5 uM CTP, 250 uM each ATP/CTP/UTP, 10 U RNase inhibitor, 1% sarkosyl), followed by Trizol RNA extraction (Invitrogen, 10296010). RNA was hydrolyzed with 0.2 N NaOH on ice for 12 min, neutralized with 0.55 M Tris-HCl pH 6.8, and buffer-exchanged with Bio-Rad P30 columns (Bio-Rad, 7326250). Biotin-labeled RNA was enriched using Dynabeads MyOne Streptavidin C1 beads, treated with RNA 5' Pyrophosphohydrolase (New England Biolabs, M0356S) and T4 Polynucleotide Kinase (New England Biolabs, M0201S), and sequentially ligated to 5' and 3' RNA adaptors by T4 RNA ligase (New England Biolabs, M0204S) with Streptavidin bead enrichments after each ligation. Reverse transcription was performed using Maxima H Minus Reverse Transcriptase (ThermoFisher Scientific, EP0751) and followed by PCR with indexed Illumina adaptors (8 cycles). 200-500 bp products were selected by SPRIselect beads (Beckman Coulter, B23318) to remove adaptor dimers. Libraries were then amplified to full scale with cycle number determined by test PCR, SPRIselect beads-purified, and sequenced via Illumina paired-end 150-bp sequencing. Reads were aligned to the mm9 genome, and duplicate reads were removed using samtools markdup. Libraries were strand-separated based on alignment

orientation and normalized to reads per million. Gene expression was quantified using the HTSeq tool[96] by counting reads aligned to gene exons, and results were presented as the ratio of mutant to WT within the same experiment.

## Antibodies
Detailed information of all antibodies used in this study is provided in Supplementary Table 2.

## Statistical analysis
Statistical analyses were performed using GraphPad Prism9 via unpaired, two-tailed Student's t test, except for Supplementary Fig. 7 f, in which P value was calculated by the Kolmogorov-Smirnov test. P values are shown as exact values, and when P ≥ 0.05, it is also labeled as ns (non-significant).

## Reporting summary
Further information on research design is available in the Nature Portfolio Reporting Summary linked to this article.

## Data availability
All data are included in the Supplementary Information or available from the authors, as are unique reagents used in this Article. The raw numbers for charts and graphs are available in the Source Data file whenever possible. High-throughput sequencing data generated in this study have been deposited at the Gene Expression Omnibus (GEO) database. HTGTS-V(D)J-Seq data is under accession code GSE287940. 3C-HTGTS data is under accession code GSE287937. ChIP-Seq data is under accession code GSE287935. PRO-Seq data is under accession code GSE305325 [https://www.ncbi.nlm.nih.gov/geo/query/acc.cgi?acc=GSE305325]. Mass spectrometry data generated in this study have been deposited at the Proteomics Identification (PRIDE) database under accession code PXD058006. E2A ChIP-Seq was extracted from a previously published GEO accession code GSM546523. FACS data was deposited to the Dryad Data Submission [https://doi.org/10.5061/dryad.2z34tmq0n]. Source data are provided with this paper.

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

## Acknowledgements

We thank the members of Zhang and Wang labs for valuable discussions. This work was supported by NIH Grants R01AI155775 (to Y.Z.), R35GM153479 (to L.S.) and R01CA247863 (to L.S.), NSF Grants DMS-2152011 and DBI-1942143 (to J.W.) and MSU Strategic Partnership Grant (to J.W. and Y.Z.). The flow cytometry analysis was performed with the Attune CytPix supported by the Equipment Grant 2022–70410-38419 from the U.S. Department of Agriculture (USDA), National Institute of Food and Agriculture (NIFA).

## Author contributions

J.W. and Y.Z. designed and supervised the project. E.B. and B.B.-R. performed most of the experiments. F.C. performed part of the ChIP-Seq and 3C-HTGTS experiments. K.H. generated the dCas9-SunTag block-ade line. J.C. generated the Vκ6-17 inversion line. J.A.C.-R. and L.S. performed and analyzed the Label-Free qMS experiments. X.Y. and J.Y. performed bioinformatic analysis. E.B., J.W., and Y.Z. summarized the data. J.W. and Y.Z. wrote the manuscript.

## Competing interests

The authors declare no competing interests.
