## [Transparent Peer Review file · Nature Communications]

CTCF couples long-range loop extrusion and diffusion to mediate a diverse Igk repertoire

Corresponding Author: Dr Yu Zhang

Version 0:

Reviewer comments:

Reviewer #1

(Remarks to the Author)

This paper tackles a fundamental mechanistic question – what are the forces that organize the immense length of the immunoglobulin loci such that the VDJ recombination machinery can operate effectively.

The study focuses on chromosome organization of Jk and Vk gene segments (over a 3.2 Mb region)

They note” Loop extrusion and diffusion are considered two key mechanisms underlying genome folding. While both processes are thought to facilitate Igk recombination, their cooperative mechanisms remain unclear. Our study demonstrates CTCF as a key regulator coupling loop extrusion and diffusion to enhance Vk recombination “

MAIN findings

* CTCF's N-terminus promotes long-range loop extrusion important for distal Vk usage by stabilizing cohesin against Wapl release.

* CTCF's N-terminus promotes loop barrier formation crucial for the joining of inversional V segments by blocking direct extrusion between Jk s and Vk s, thereby enabling diffusion.

* In CTCF N-terminal mutants, defects in inversional Vk joining were not corrected by Wapl depletion but largely rescued by a targeted dCas9-blockade mimicking the CTCF barrier.

This is a technically impressive dissection of the role of CTCF and Wapl in the regulation of deletion vs inversional Vk joining and the conformation of the kappa locus.

For completeness, they should reference <https://pmc.ncbi.nlm.nih.gov/articles/PMC10917333/> and mention the role of the E34 enhancer

(review by Rachel Gerstein, UMASS Chan Medical School)

Reviewer #2

(Remarks to the Author)

This manuscript represents an outstanding follow-up to prior work by the senior author addressing the mechanisms by which Vk gene segments are brought into proximity of Jk gene segments for Igk recombination. The prior work had demonstrated that RAG cannot scan from Jk to Vk RSSs; rather, loop extrusion within the Vk array followed diffusion-based RSS synapsis is key to Vk-Jk recombination. Here the authors use a v-Abl pro-B system to analyze mechanisms by which the CTCF N-terminus regulates these processes. Mutation of the CTCF N-terminus caused increased usage of proximal V segments in deletional orientation while suppressing the use of distant V segments of either orientation. The former was inferred to result from reduced Cer-Sis barrier formation, and the latter from reduced long-range interactions with Cer. Defective looping was associated with substantially reduced chromatin-associated SMC3, and reduced SMC3-CTCF but increased SMC3-Wapl interactions, leading to the conclusion that increased cohesin release by Wapl in the mutant impairs long-range loop extrusion. Consistent with this, in CTCF Nm, Wapl degradation led to increased long-range interactions/loop extrusion across the Vk segments, and increased rearrangement of distant V gene segments. But in contrast to Wapl depletion with WT CTCF, rearrangement increased only for those Vk segments that rearrange by deletion, implicating impaired Cer-Sis barrier function. Finally, the authors used an elegant dCas9 blockade system to prove that it is indeed the loss of Cer-Sis barrier function in CTCF Nm that drives the locus towards deletional rearrangement, and that blockade of loop extrusion from Jks to Vks is critical for diffusion to be the dominant mechanism for Vk-Jk recombination. It is concluded that the CTCF

N-terminus facilitates Igk repertoire formation by enforcing Cer-Sis barrier function and by facilitating cohesin-mediated loop extrusion across the Vk segments. This work is technically sound and represents a substantial advance for the field. The manuscript is very well written.

1. The authors acknowledge that CTCF-Nm results in mild reduction of CTCF-binding genome-wide, while the occupancy at Igk locus is significantly reduced. In that case, how do they conclude the resulting phenotype is not simply due to reduced CTCF binding, rather than its impaired interaction with cohesin? It will be informative to show actual CTCF binding peaks, instead of "called" CTCF sites in the highlighted regions in the figures. Also, binding at some IgK CTCF sites seems to be less affected by the mutation than others. For the Vks with no difference in recombination frequencies (e.g. Vks 8-30, 8-21, 8-16), it will be interesting to know the status of nearby CTCF binding.

2. It may be interesting to compare CBE sequences of sites that lose or retain CTCF binding in CTCF Nm, both within Igk and genome-wide.

3. Can the authors confirm that there is minimal effect of CTCF-Nm on CTCF binding at Cer and Sis?

4. In Li et al., Nature, 2020, CTCF YDF mutation caused differential gene expression of >2000 genes in HAP1 cells. Have the authors studied the changes in gene expression in their system? Are there changes in the expression of any genes known to affect loop extrusion and diffusion?

Minor:

line 231 – CTCF deficient should be CTCF Nm

line 251 – Can the authors comment on the fact that interactions with the BS region do not appear to increase with dCas9-blockade?

Lines 261,2 – Mention of increased cohesin processivity, without further explanation or discussion, is a bit obtuse, and is perhaps not helpful.

Lines 318-321 – Since the authors note that secondary rearrangements occur after the Cer-Sis barrier is removed, it may be helpful to note that although these rearrangements should occur primarily by deletion, they can involve V segments that were initially in the wrong orientation since their orientation can be flipped by inversional primary rearrangement.

Reviewer #3

(Remarks to the Author)

The question of how numerous V gene segments, scattered over megabase-scale distances, successfully participate in V(D)J recombination to generate a diverse antigen receptor repertoire has been a central issue in the field for decades. While RAG chromatin scanning was definitively shown by this group to drive most D-to-J_H recombination, the mechanism for V_k-to-J_k joining is more complex. In the Igk locus, RAG utilizes V_k-RSSs from both deletional- and inversional-oriented gene clusters, inconsistent with a purely unidirectional linear scanning model.

A recent study (Zhang et al., PMID: 38811728) proposed that Igk recombination is uniquely "diffusion-friendly," relying on strong RSSs and limited extrusion distances to support both deletional and inversional joining. In contrast, IgH recombination is more dependent on long-range loop extrusion and weaker RSSs, thereby biasing strongly toward deletional outcomes.

This manuscript makes a significant mechanistic contribution by using an established CTCF N-terminal (CTCF-Nm) mutation to dissect how the architectural protein CTCF governs Igk recombination. The authors convincingly demonstrate that the CTCF-Nm mutation reduces both distal V_k usage and long-range interactions with the Cer/Sis elements, while also decreasing inversional recombination events.

Through use of Wapl-degron and dCas9-SunTag tethering, the authors provide strong evidence that CTCF has dual roles:

1. Cohesin stabilization: CTCF's N-terminus resists Wapl-mediated unloading, facilitating long-range loop extrusion to engage distal V_ks. This aligns with Zhang et al.'s observations on local scanning constraints.
2. Extrusion barrier formation: CTCF halts cohesin progression at specific boundaries, creating a chromosome architecture and topological window that permits diffusion-mediated inversional pairing.

By establishing these two distinct yet complementary roles, the authors extend the "extrusion + diffusion" model of Igk recombination. This work represents a major contribution to our understanding of how genome architecture, recombination signal strength, and chromatin dynamics are integrated to regulate immune diversity.

The only major comment is that the paper needs to be revised to make the hypotheses, findings, and conclusions clearer and easy to conceptualize.

Minor Points for Clarification or Revision:

1. Line 27: "Defects in inversional V joining were not corrected by Wapl."

→ Consider revising to: "...were not fully restored by Wapl depletion" for better accuracy.

2. Line 29–30: "CTCF regulates immune diversity via diverse mechanisms..."

→ It would strengthen the statement to explicitly mention the two mechanistic roles uncovered in this study: (1) stabilizing loop extrusion via cohesin retention and (2) acting as a chromatin barrier that enables diffusion-based inversional joining.

3. Line 99: The authors observe a 3.8 fold reduction in recombination with CTCF-Nm. They should confirm that RAG1 expression levels are comparable between the WT and CTCF-Nm cells (assuming it is feasible to perform RAG1 westerns). Normally this would not be a concern but the experiments are being performed in a cell line that also expresses substantial levels of RAG1 D708A. RAG1 forms very tight, stable dimers and WT and D708A RAG1 will dimerize, soaking up a

significant fraction of the WT RAG1 into heterodimers with only one functional active site. As a result, even small differences in WT RAG1 expression could result in large differences in levels of fully active WT-WT homodimers. The authors should acknowledge this issue somewhere in the manuscript.

4. Line 135-137: the way the conclusion is stated is incomplete and a bit misleading. First, the clearest change in the recombination pattern with CTCF-Nm is the large increase usage of proximal forward Vk segments, so one obvious function of CTCF is suppression of high level usage of these segments—and yet the authors do not mention this function. Second, the claim of “mediating” long range recombination goes beyond the data at this point in the manuscript. The decrease in long range recombination could be entirely due to the increase in proximal recombination, in which case CTCF is not positively supporting long range recombination (as implied by “mediating”), but “supports” or “enables” long range recombination by suppressing short-range.

5. Line 171: “lack” is too strong. “decreased” would be appropriate.

6. Line 216: It is not clear what is “remarkable” about this finding. Data from Fig. 1 had already shown that inversional recombination was reduced with CTCF-Nm and retained with WT CTCF. What is remarkable about the fact that inversional recombination is unchanged by loss of WAPL with WT CTCF? Please explain more clearly or rewrite.

7. Line 222: please cite the relevant figure panels.

8. Line 259: Figure 6f does not allow the reader to compare the recombination pattern in the blockage line to that of WT cells. A figure providing a direct comparison would be helpful.

9. Clearer presentation requested: when referring to the dCas9 blockade (for example, line 28 in the abstract) it is vital that the authors make clear where the blockade is located. When I read the abstract, I inferred that the blockade was located in the Vk region, only to discover when I got to Figure 6 that the blockade was in a completely different place, between Jk and Vk. The last paragraph of the discussion is very important as it is the first place where the authors distinguish between CTCF sites and functions in Vk versus in the Jk-Vk interval (and throughout the paper, I thought that it was the CTCF sites in Vk that were the focus of the analysis, only to discover that they were not the focus at all). The authors are strongly encouraged to make it clear what they have done with the blockade and why, from the very beginning of the manuscript, and to make clear what they are focusing on and why throughout the manuscript. And see point 14 below: a well-illustrated model figure will help greatly in allowing the authors to specify the role of CTCF at Cer/Sis versus in the Vk region.

10. Figure labeling (e.g., Fig. 1b): The use of “forward Vks” and “reverse Vks” is explained in the text but is not always easy to work with in the figures. For example, in Fig. 1b, it would be helpful if the labels “forward” and “reverse” had a second label in parentheses saying “deletional” and “inversional” to align more clearly with the bar graph labels.

11. Mechanistic clarification requested: The CTCF-Nm mutation weakens the Cer–Sis boundary, allowing loop extrusion and RAG scanning into Vks. And strong RSSs in the Vk region are capable of mediating inversional joining via diffusion. Chromatin loop extrusion will bring the reverse Vk segments close to the Jk RC and yet they don't recombine efficiently, and this observation is central to the manuscript. The authors should carefully and clearly explain how their findings and model explain this observation. Again, a good graphical model will be a big help.

12. Rearrangement of weak RSSs: Do weak distal Vk RSSs or cryptic RSSs (cRSS) show increased recombination when loop extrusion remains intact but barrier function is compromised (e.g., in Wapl-AID2 CTCF-Nm cells)? Any observed cRSS recombination upstream of the Igk locus would support the model. Information regarding cRSS recombination might be available in the HTGTS data already collected.

13. Chromatin accessibility and epigenetics: Were chromatin activation marks such as H3K27ac or H3K4me3 profiled across the Vk region in CTCF-Nm mutant cells? This could help differentiate direct architectural changes from secondary effects on enhancer activity and chromatin changes associated with transcription. This would be a nice addition but is not required.

14. Model illustration requested: A graphical model illustrating deletional vs. inversional recombination routes, highlighting the roles of CTCF, cohesin, Wapl, and dCas9, would be very helpful for readers to visualize the “extrusion + diffusion” framework. In addition, it would help orient readers to have a schematic in Figure 1 showing the conceptual framework that guided the design of the experiments: that is, the RC looping to SIS and the V regions looping to CER.

Reviewer #4

(Remarks to the Author)

Version 1:

Reviewer comments:

Reviewer #2

(Remarks to the Author)

The authors have satisfactorily addressed all of my questions and comments and I am strongly in favor of publication.

I suggest one minor edit:

Lines 112-113: rewrite to “which might further indicate a negative impact on Igk recombination center activity or RAG activity by the CTCF-Nm mutation”

Reviewer #3

(Remarks to the Author)

The authors have done an excellent job addressing our comments and those of the other reviewers. The model figures are a big help in making the results understandable and the conclusions clear. The manuscript is much improved and we support publication.

Reviewer #4

(Remarks to the Author)

Point-by-point responses to reviewer comments

REVIEWER COMMENTS

Reviewer #1 (B cell development, VDJ) (Remarks to the Author):

This paper tackles a fundamental mechanistic question – what are the forces that organize the immense length of the immunoglobulin loci such that the VDJ recombination machinery can operate effectively.

The study focuses on chromosome organization of Jk and Vk gene segments (over a 3.2 Mb region)

They note” Loop extrusion and diffusion are considered two key mechanisms underlying genome folding. While both processes are thought to facilitate Igk recombination, their cooperative mechanisms remain unclear. Our study demonstrates CTCF as a key regulator coupling loop extrusion and diffusion to enhance Vk recombination “

MAIN findings

* CTCF's N-terminus promotes long-range loop extrusion important for distal Vk usage by stabilizing cohesin against Wapl release.

* CTCF's N-terminus promotes loop barrier formation crucial for the joining of inversional Vk segments by blocking direct extrusion between Jk s and Vk s, thereby enabling diffusion.

* In CTCF N-terminal mutants, defects in inversional Vk joining were not corrected by Wapl depletion but largely rescued by a targeted dCas9-blockade mimicking the CTCF barrier.

This is a technically impressive dissection of the role of CTCF and Wapl in the regulation of deletion vs inversional Vk joining and the conformation of the kappa locus.

Response: We thank the reviewer for this very positive assessment and for recognizing the ‘impressive’ strength of our experimental design and technical approach in supporting our model.

For completeness, they should

reference <https://pmc.ncbi.nlm.nih.gov/articles/PMC10917333/> and mention the role of the E34 enhancer

(review by Rachel Gerstein, UMASS Chan Medical School)

Response: This is a great suggestion, we have now added this reference of E34 and described its relevant function in the Introduction.

Reviewer #2 (VDJ recombination, epigenetic regulation, DNA repair) (Remarks to the Author):

This manuscript represents an outstanding follow-up to prior work by the senior author addressing the mechanisms by which Vk gene segments are brought into proximity of Jk gene segments for Igk recombination. The prior work had demonstrated that RAG cannot scan from Jk to Vk RSSs; rather, loop extrusion within the Vk array followed diffusion-based RSS synapsis is key to Vk-Jk recombination. Here the authors use a v-Abl pro-B system to analyze mechanisms by which the CTCF N-terminus regulates these processes. Mutation of the CTCF N-terminus caused increased usage of proximal V segments in deletional orientation while suppressing the use of distant V segments of either orientation. The former was inferred to result from reduced Cer-Sis barrier formation, and the latter from reduced long-range interactions with Cer. Defective looping was associated with substantially reduced chromatin-associated SMC3, and reduced SMC3-CTCF but increased SMC3-Wapl interactions, leading to the conclusion that increased cohesin release by Wapl in the mutant impairs long-range loop extrusion. Consistent with this, in CTCF Nm, Wapl degradation led to increased long-range interactions/loop extrusion across the Vk segments, and increased rearrangement of distant V gene segments. But in contrast to Wapl depletion with WT CTCF, rearrangement increased only for those Vk segments that rearrange by deletion, implicating impaired Cer-Sis barrier function. Finally, the authors used an elegant dCas9 blockade system to prove that it is indeed the loss of Cer-Sis barrier function in CTCF Nm that drives the locus towards deletional rearrangement, and that blockade of loop extrusion from Jks to Vks is critical for diffusion to be the dominant mechanism for Vk-Jk recombination. It is concluded that the CTCF N-terminus facilitates Igk repertoire formation by enforcing Cer-Sis barrier function and by facilitating cohesin-mediated loop extrusion across the Vk segments. This work is technically sound and represents a substantial advance for the field. The manuscript is very well written.

Response: We thank the reviewer for this strongly positive review which clearly points out the high quality of our experiments and the high impact of our studies for the field. We appreciate that the review found our work to be 'outstanding', 'technically sound' and 'represents a substantial advance for the field'.

We also thank the reviewer for the compliments on our writing.

1. The authors acknowledge that CTCF-Nm results in mild reduction of CTCF-binding genome-wide, while the occupancy at Igk locus is significantly reduced. In that case, how do they conclude the resulting phenotype is not simply due to reduced CTCF binding, rather than its impaired interaction with cohesin? It will be informative to show actual CTCF binding peaks, instead of "called" CTCF sites in the highlighted regions in the figures. Also, binding at some IgK CTCF sites seems to be less affected by the mutation than others. For the Vks with no difference in recombination frequencies (e.g. Vks 8-30, 8-21, 8-16), it will be interesting to know the status of nearby CTCF binding.

Response: This is an excellent question. Several lines of evidence strongly suggest that the CTCF-Nm phenotype results from defective N-terminal-dependent stabilization of cohesin, rather than reduced CTCF binding per se: 1. Distal Vks within regions of relatively preserved CTCF binding still showed significantly reduced usage, indicating that retained CTCF was non-functional (Supplementary Fig. 5c). 2. WAPL depletion,

which enhances cohesin processivity without affecting CTCF binding, rescued distal deletional joining in the CTCF-Nm mutants (Fig. 4e). 3. Despite only mild reductions in binding at the Cer-Sis region, long-range interactions with distal Vks, including those mediated by E2A, were substantially diminished (Supplementary Fig. 5b; Fig. 3a). 4. Targeted dCas9 blockade near Cer-Sis, which is unlikely to affect CTCF binding, strongly rescued recombination across the locus (Fig. 6f). We have now incorporated these points into the Discussion section (highlighted in yellow).

We also appreciate the reviewer's suggestion to overlay Vk segments with interaction profiles and actual CTCF ChIP-Seq signals, particularly for CTCF sites less affected by the mutation and Vks whose usage was less impacted. To fully address this, we have included a zoomed-in view of a 400-kb distal Vk region containing CTCF sites relatively insensitive to CTCF-Nm, presented after the introduction of the CTCF ChIP-Seq data; Vk segments in this region exhibited similar decreases in usage as other distal Vks (Supplementary Fig. 5c). This indicates that, although differential binding likely reflects intrinsic CTCF site strength, retained binding in CTCF-Nm mutants may be non-functional due to the N-terminal mutation.

Additionally, we included zoomed-in views of a proximal Vk region containing multiple inversional Vks showing little change in usage, and found that these Vks lack adjacent CTCF binding sites (Supplementary Fig. 5d). This indicates that retained CTCF binding is not responsible for their preserved usage.

2. It may be interesting to compare CBE sequences of sites that lose or retain CTCF binding in CTCF Nm, both within Igk and genome-wide.

Response: This is a good point. We have now included a comparative analysis of CTCF sites that retain or lose binding in CTCF-Nm mutant cells, both genome-wide and within the Ig κ locus (Supplementary Fig. 5e). The analysis encompassed the 19-bp core motif plus 16 bp upstream and 18 bp downstream sequences, but revealed no obvious differences in DNA sequence composition. This may suggest that factors beyond the core motif and its immediate flanking sequences contribute to the N-terminal-dependent binding sensitivity, warranting further investigation in future studies.

3. Can the authors confirm that there is minimal effect of CTCF-Nm on CTCF binding at Cer and Sis?

Response: We agree with the reviewer that this is an important point to highlight. We have now added zoomed-in views of CTCF binding at the Cer-Sis region in Supplementary Fig. 5b, which demonstrated that unlike many V κ CTCF sites, CTCF binding at Cer and Sis is only mildly impacted in CTCF-Nm cells. This observation is now cited in the revised Results section.

4. In Li et al., Nature, 2020, CTCF YDF mutation caused differential gene expression of >2000 genes in HAP1 cells. Have the authors studied the changes in gene expression in their system? Are there changes in the expression of any genes known to affect loop extrusion and diffusion?

Response: This is a great question. We have now performed and added PRO-Seq data for WT and CTCF-Nm mutants from three independent biological replicates.

Comparative gene expression analysis showed that the CTCF-Nm mutation does not significantly impact the expression of key factors known or hypothesized to be involved in loop extrusion or diffusion (Supplementary Fig. 6a).

Minor:

line 231 – CTCF deficient should be CTCF Nm

Response: Thanks for pointing this out, we have now corrected it.

line 251 – Can the authors comment on the fact that interactions with the BS region do not appear to increase with dCas9-blockade?

Response: This is a great question. We believe the lack of interaction peaks at the BS region is due to the dCas9-SunTag complex physically obstructing NlaIII digestion during 3C-HTGTS library preparation, thereby preventing detection of these fragments. The BS region contains four NlaIII recognition sites (CATG), all within 5 bp of BS, making them particularly susceptible to steric hindrance by dCas9 binding. This explanation is consistent with previous reports that dCas9 occupancy can block restriction digestion (Saifaldeen et al., 2021, PMID: 33876957). We have illustrated the positions of the NlaIII sites relative to BSs in Supplementary Fig. 9a and now include this interpretation in the Results section.

Lines 261,2 – Mention of increased cohesin processivity, without further explanation or discussion, is a bit obtuse, and is perhaps not helpful.

Response: Thanks for pointing this out, we agree with the reviewer and have now deleted this sentence.

Lines 318-321 – Since the authors note that secondary rearrangements occur after the Cer-Sis barrier is removed, it may be helpful to note that although these rearrangements should occur primarily by deletion, they can involve V segments that were initially in the wrong orientation since their orientation can be flipped by inversional primary rearrangement.

Response: We have now edited the Discussion to include this point.

Reviewer #3 (VDJ recombination, DNA repair, SHM) (Remarks to the Author):

The question of how numerous V gene segments, scattered over megabase-scale distances, successfully participate in V(D)J recombination to generate a diverse antigen receptor repertoire has been a central issue in the field for decades. While RAG chromatin scanning was definitively shown by this group to drive most D-to-J_H recombination, the mechanism for V_κ-to-J_κ joining is more complex. In the Ig_κ locus, RAG utilizes V_κ-RSSs from both deletional- and inversional-oriented gene clusters, inconsistent with a purely unidirectional linear scanning model.

A recent study (Zhang et al., PMID: 38811728) proposed that Ig_κ recombination is uniquely "diffusion-friendly," relying on strong RSSs and limited extrusion distances to support both deletional and inversional joining. In contrast, Ig_H recombination is more dependent on long-range loop extrusion and weaker RSSs, thereby biasing strongly toward deletional outcomes.

This manuscript makes a significant mechanistic contribution by using an established

CTCF N-terminal (CTCF-Nm) mutation to dissect how the architectural protein CTCF governs Igk recombination. The authors convincingly demonstrate that the CTCF-Nm mutation reduces both distal Vk usage and long-range interactions with the Cer/Sis elements, while also decreasing inversional recombination events.

Through use of Wapl-degron and dCas9-SunTag tethering, the authors provide strong evidence that CTCF has dual roles:

1. Cohesin stabilization: CTCF's N-terminus resists Wapl-mediated unloading, facilitating long-range loop extrusion to engage distal Vks. This aligns with Zhang et al.'s observations on local scanning constraints.

2. Extrusion barrier formation: CTCF halts cohesin progression at specific boundaries, creating a chromosome architecture and topological window that permits diffusion-mediated inversional pairing.

By establishing these two distinct yet complementary roles, the authors extend the "extrusion + diffusion" model of Igk recombination. This work represents a major contribution to our understanding of how genome architecture, recombination signal strength, and chromatin dynamics are integrated to regulate immune diversity.

Response: We thank the reviewer for this highly positive review, highlighting that our evidence strongly supports our proposed model and represents 'a significant mechanistic advance' in the field.

The only major comment is that the paper needs to be revised to make the hypotheses, findings, and conclusions clearer and easy to conceptualize.

Response: Thanks for this valuable feedback. In response, we have revised the manuscript to improve the clarity and presentation of the hypotheses, findings, and conclusions as suggested. Importantly, we have added two model figures, one main (Fig. 7) and one supplementary (Supplementary Fig. 10) to help explaining our findings.

Minor Points for Clarification or Revision:

1. Line 27: "Defects in inversional V joining were not corrected by Wapl."

→ Consider revising to: "...were not fully restored by Wapl depletion" for better accuracy.

Response: Thanks for the suggestion. We have revised the wording, and now the sentence reads as "Defects in inversional V joining were not *restored* by WAPL". We opted not to use "fully," as it could suggest a partial rescue, whereas our data indicate that Wapl depletion did not restore the inversional joining defects to any extent.

2. Line 29–30: "CTCF regulates immune diversity via diverse mechanisms..."

→ It would strengthen the statement to explicitly mention the two mechanistic roles uncovered in this study: (1) stabilizing loop extrusion via cohesin retention and (2) acting as a chromatin barrier that enables diffusion-based inversional joining.

Response: Thanks for the suggestion. We have revised this sentence to "*These findings highlight how CTCF coordinates distinct genome-folding mechanisms through its dual roles in cohesin stabilization and extrusion barrier formation to ensure efficient Vk recombination, providing insights into antibody diversification and genome regulation more broadly.*" We opted not to restate the full details of the specific roles of CTCF in this final sentence because they are already described earlier in the abstract: "We show

that CTCF's N-terminus promotes long-range loop extrusion important for distal Vk usage by stabilizing cohesin against Wapl release,” and “Furthermore, CTCF's N-terminus promotes loop barrier formation crucial for the joining of inversional Vk segments.”

3. Line 99: The authors observe a 3.8 fold reduction in recombination with CTCF-Nm. They should confirm that RAG1 expression levels are comparable between the WT and CTCF-Nm cells (assuming it is feasible to perform RAG1 westerns). Normally this would not be a concern but the experiments are being performed in a cell line that also expresses substantial levels of RAG1 D708A. RAG1 forms very tight, stable dimers and WT and D708A RAG1 will dimerize, soaking up a significant fraction of the WT RAG1 into heterodimers with only one functional active site. As a result, even small differences in WT RAG1 expression could result in large differences in levels of fully active WT-WT homodimers. The authors should acknowledge this issue somewhere in the manuscript.

Response: Thanks for this insightful suggestion. We have now performed PRO-Seq analysis and found that, while the CTCF-Nm mutation has little impact on other key factors involved in V(D)J recombination, it increases the expression of RAG1 and RAG2 (Supplementary Fig. 6c). Western blot analysis, as suggested, further confirmed enrichment of RAG1 protein (Supplementary Fig. 6d). One potential explanation is that RAG1 and RAG2 are expressed as transgenes in these cells and might be more susceptible to the effects of the CTCF-Nm mutation. We have included these results and acknowledged this issue in the manuscript.

Of note, PRO-Seq analysis also showed a significant impact of CTCF-Nm on transcriptional activity at the recombination center (RC), but not at the Vk region (Supplementary Fig. 6b), consistent with our prior hypothesis that CTCF-Nm negatively impacts RC activity. Future studies will help to determine the relative contributions of these mechanisms to the overall reduction in Igk recombination.

4. Line 135-137: the way the conclusion is stated is incomplete and a bit misleading. First, the clearest change in the recombination pattern with CTCF-Nm is the large increase usage of proximal forward Vk segments, so one obvious function of CTCF is suppression of high level usage of these segments—and yet the authors do not mention this function. Second, the claim of “mediating” long range recombination goes beyond the data at this point in the manuscript. The decrease in long range recombination could be entirely due to the increase in proximal recombination, in which case CTCF is not positively supporting long range recombination (as implied by “mediating”), but “supports” or “enables” long range recombination by suppressing short-range.

Response: Thanks for pointing this out. We have revised the text accordingly where we first introduced the Igk recombination defect phenotype “However, in the CTCF-Nm mutants, there was a considerable change in the pattern of Vk utilization, with significantly decreased middle/distal Vk usage accompanied by a substantial increase in the frequency of proximal Vk usage (Fig. 1b, left panel). This suggests the CTCF N-terminus plays an important role in *supporting* long-range Vk joining, *though it remains unclear at this point whether this effect occurs indirectly by suppressing proximal Vk joining or directly by promoting distal Vk joining.*”, Additionally, we replaced “mediating”

with “supporting” as suggested in the reference sentence “Collectively, the HTGTS-V(D)J-Seq analysis indicated two major roles of the CTCF-N terminus in Igk recombination: (1) *supporting* long-range Vk usage, and (2) promoting the utilization of inversional Vks. We then sought out to explore the mechanistic basis.”

5. Line 171: “lack” is too strong. “decreased” would be appropriate.

Response: We have now replaced “lack” with “*decreased.*”

6. Line 216: It is not clear what is “remarkable” about this finding. Data from Fig. 1 had already shown that inversional recombination was reduced with CTCF-Nm and retained with WT CTCF. What is remarkable about the fact that inversional recombination is unchanged by loss of WAPL with WT CTCF? Please explain more clearly or rewrite.

Response: We have now replaced “Remarkably, however” with “*On the other hand.*”

7. Line 222: please cite the relevant figure panels.

Response: We have now added the relevant figure citation.

8. Line 259: Figure 6f does not allow the reader to compare the recombination pattern in the blockage line to that of WT cells. A figure providing a direct comparison would be helpful.

Response: We have now added a new figure panel (Supplementary Fig. 9b) to provide a direct comparison of Vk utilization frequency between the dCas9-blockade and WT lines.

9. Clearer presentation requested: when referring to the dCas9 blockade (for example, line 28 in the abstract) it is vital that the authors make clear where the blockade is located. When I read the abstract, I inferred that the blockade was located in the Vk region, only to discover when I got to Figure 6 that the blockade was in a completely different place, between Jk and Vk. The last paragraph of the discussion is very important as it is the first place where the authors distinguish between CTCF sites and functions in Vk versus in the Jk-Vk interval (and throughout the paper, I thought that it was the CTCF sites in Vk that were the focus of the analysis, only to discover that they were not the focus at all). The authors are strongly encouraged to make it clear what they have done with the blockade and why, from the very beginning of the manuscript, and to make clear what they are focusing on and why throughout the manuscript. And see point 14 below: a well-illustrated model figure will help greatly in allowing the authors to specify the role of CTCF at Cer/Sis versus in the Vk region.

Response: We have made the following revisions to clarify the design and location of the dCas9 blockade, as well as to distinguish the potential functional differences between CTCF sites at the Igk Cer-Sis region and other Vk CTCF sites:

1. Abstract: We now explicitly indicate the location of the dCas9 blockade: “In CTCF N-terminal mutants, defects in inversional Vk joining were not restored by WAPL depletion but largely rescued by a dCas9-blockade *targeted to the Vk-Jk intergenic region*, mimicking the CTCF barrier.”

2. Introduction: further highlighted the role of Cer-Sis: *Importantly, Vks are insulated from the recombination center by the Cer and Sis elements interacting with the Vk region and recombination center, respectively*⁴⁷. Each element contains two CTCF sites, with those in Cer oriented toward Vks and those in Sis oriented toward the recombination center, together supporting a balanced Vk repertoire^{54, 55} in a manner dependent on the orientation of these CTCF sites⁵⁶.

3. Results, CTCF ChIP-Seq profiles: We have now added a zoomed-in profile of Igk Cer-Sis region, showing that unlike many Vk CTCF sites, CTCF binding at Cer-Sis is largely retained: “We also noted that while the mutation only mildly reduced CTCF binding genome-wide *and at the Igk Cer-Sis elements*, binding at Vk locus was significantly decreased at many loci, suggesting many Vk CTCF sites have relatively weak affinity that is dependent on a functional N-terminus of CTCF (*Supplementary Fig. 5b*)”

4. Results, dCas9 blockade experiments: We have added a sentence explaining the rationale for targeting the blockade to the Vk–Jk intergenic region: “The CTCF barriers likely enable diffusion by preventing direct extrusion between Jks and Vks, a process that would otherwise inhibit inversional RAG recombination. *This hypothesis predicts that extrusion barriers at the Vk-Jk intergenic region are of critical importance.*”

5. Results, ending sentence of the dCas9-blockade section: We have emphasized the greater importance of the Vk-Jk intergenic barrier compared with other locus-wide barriers: “Collectively, these findings provided strong evidence that extrusion barriers, *particularly at the Vk-Jk intergenic region*, are crucial for inversional Vk joining via diffusion.”

6. Discussion/Model figure: Importantly, we have added two model figures, one main and one supplementary, to more clearly illustrate our findings. We have added a schematic model in Fig. 7 to highlight the role of Cer-Sis (Fig. 7a) and the impact of the dCas9 blockade (Fig. 7c). In addition, we have included a model in Supplementary Fig. 10 to illustrate the potential roles of Vk CTCF sites during primary and secondary rearrangement, respectively.

10. Figure labeling (e.g., Fig. 1b): The use of “forward Vks” and “reverse Vks” is explained in the text but is not always easy to work with in the figures. For example, in Fig. 1b, it would be helpful if the labels “forward” and “reverse” had a second label in parentheses saying “deletional” and “inversional” to align more clearly with the bar graph labels.

Response: Thanks for the suggestion. We have now added 'Del' (standing for deletional) and 'Inv' (standing for inversional) in parentheses throughout the main and supplementary figures.

11. Mechanistic clarification requested: The CTCF-Nm mutation weakens the Cer–Sis boundary, allowing loop extrusion and RAG scanning into Vks. And strong RSSs in the Vk region are capable of mediating inversional joining via diffusion. Chromatin loop extrusion will bring the reverse Vk segments close to the Jk RC and yet they don’t recombine efficiently, and this observation is central to the manuscript. The authors should carefully and clearly explain how their findings and model explain this observation. Again, a good graphical model will be a big help.

Response: We fully agree that a clear explanation and graphical model are important. To aid clarity, we have now added a new model (Figure 7) summarizing our findings. We believe this model, together with the following discussion, highlights the central observation mentioned above, which showed that efficient inversional RSS joining requires the barrier function of CTCF.

“Our findings also highlight that long-range loop extrusion is necessary but not sufficient for recombination of most inversional Vks, which also depend on CTCF’s role as an extrusion barrier. This contrasts with most deletional Vk recombination that can occur efficiently without CTCF barriers. Accordingly, WAPL depletion in CTCF-Nm cells restores distal deletional Vk joining but does not rescue inversional Vk joining (Fig. 4e, Fig. 7b). Moreover, a striking difference was observed in inversional Vk joining frequency between WAPL-depleted CTCF-Nm cells and WAPL-depleted only cells retaining CTCF barrier function (Fig. 5b).

Although long suspected, our study provided the direct evidence that mechanistically, the ability of CTCF barriers to block direct extrusion between Jks and Vks is essential for inversional Vk joining as loop extrusion strongly favors deletional joins. In this regard, inversion of the proximal Vk region in CTCF-Nm cells compromising CTCF barriers largely reversed the recombination strength of the contained Vks based on their RSS orientations (Fig. 2b-d). Moreover, introducing a CTCF-barrier mimicking dCas9 blockade targeting the Vk-to-Jk intergenic region significantly enhances inversional Vk joining across the Igk locus in the CTCF-Nm background (Fig. 6f).”

12. Rearrangement of weak RSSs: Do weak distal Vk RSSs or cryptic RSSs (cRSS) show increased recombination when loop extrusion remains intact but barrier function is compromised (e.g., in Wapl-AID2 CTCF-Nm cells)? Any observed cRSS recombination upstream of the Igk locus would support the model. Information regarding cRSS recombination might be available in the HTGTS data already collected.

Response: As suggested, we examined rearrangement of weak distal deletional Vks (the 11 least-used prior to Wapl depletion) compared with strong distal deletional Vks (the 11 most-used prior to Wapl depletion) following Wapl depletion in CTCF-Nm cells. No significant differences were observed, consistent with a model in which the changes reflect increased RAG scanning rather than selective targeting of certain strong Vk-RSSs. This analysis has been added to the Results (Supplementary Fig. 7e). We also examined cRSS usage, but these events were detected only at very low frequencies in Wapl-depleted CTCF-Nm cells and showed minimal recurrence across replicates. Given this low signal, which is consistent with limited RC activity and possibly rapid bypass of potential cRSS sites during extended loop extrusion in the absence of barriers in these cells, further detailed analysis was prevented.

13. Chromatin accessibility and epigenetics: Were chromatin activation marks such as H3K27ac or H3K4me3 profiled across the Vk region in CTCF-Nm mutant cells? This could help differentiate direct architectural changes from secondary effects on enhancer activity and chromatin changes associated with transcription. This would be a nice addition but is not required.

Response: Thanks for the suggestion. Although we do not currently have H3K27ac or H3K4me3 ChIP-Seq data, we have now performed PRO-Seq analysis (Supplementary Fig. 6b) that directly measures transcription activity. The results show that germline transcription across the Vk region is largely maintained in CTCF-Nm mutant cells, indicating that altered transcriptional/chromatin accessibility across the Vk region does not explain the observed Igk recombination defects in long-range Vk usage and inversional joining. This new analysis has been incorporated into the Results section.

14. Model illustration requested: A graphical model illustrating deletional vs. inversional recombination routes, highlighting the roles of CTCF, cohesin, Wapl, and dCas9, would be very helpful for readers to visualize the "extrusion + diffusion" framework. In addition, it would help orient readers to have a schematic in Figure 1 showing the conceptual framework that guided the design of the experiments: that is, the RC looping to SIS and the V regions looping to CER.

Response: Thanks for this excellent suggestion. We have now added a model in the main Figure 7 to illustrate the roles of Cer-Sis, cohesin, Wapl, and dCas9. In addition, Supplementary Figure 10 has been included to illustrate the potential roles of CTCF sites in Vk region.

We also appreciate the suggestion of highlighting the findings in a previous elegant study that the RC looping to SIS and the V regions looping to CER. We have incorporated the corresponding text in both Introduction to provide relevant background and Discussion for the clarity of the model.

Introduction: *“Importantly, Vks are insulated from the recombination center by the Cer and Sis elements interacting with the Vk region and recombination center, respectively⁴⁷. Each element contains two CTCF sites, with those in Cer oriented toward Vks and those in Sis oriented toward the recombination center, together supporting a balanced Vk repertoire^{54, 55} in a manner dependent on the orientation of these CTCF sites⁵⁶”.*

Discussion: “In this regard, loop extrusion likely brings the widely separated Vks and Jks into close proximity, within a subdiffusive range, at which point diffusion takes over. *This aligns with a recent elegant study showing that the Vk region loops to the Cer element and the Jk recombination center loops to the Sis element⁴⁷, a configuration likely enabled by the specific orientation organization of CTCF sites within these elements, with their N-termini facing Vks and Jks, respectively (Fig. 7a).* “

Reviewer #4 (EC) (Remarks to the Author):

Response: We thank the reviewer for the careful and insightful evaluation of our work.

Point-by-point responses to reviewer comments

REVIEWERS' COMMENTS

Reviewer #2 (Remarks to the Author):

The authors have satisfactorily addressed all of my questions and comments and I am strongly in favor of publication.

Response: We thank the reviewer for this strongly positive assessment of our revision.

I suggest one minor edit:

Lines 112-13: rewrite to "which might further indicate a negative impact on Igk recombination center activity or RAG activity by the CTCF-Nm mutation"

Response: Thanks. We have edited the text as suggested.

Reviewer #3 (Remarks to the Author):

The authors have done an excellent job addressing our comments and those of the other reviewers. The model figures are a big help in making the results understandable and the conclusions clear. The manuscript is much improved and we support publication.

Response: We thank the reviewer for this highly positive assessment of our revision.

Reviewer #4 (Remarks to the Author):

Response: We thank the reviewer for the careful evaluation of our work.